

# Validation of a coupled atmospheric-aeroelastic model system for wind turbine power and load calculations

Sonja Krüger[1], Gerald Steinfeld[1], Martin Kraft[1], and Laura J. Lukassen[1]

[1]ForWind, Institute of Physics, Carl von Ossietzky University Oldenburg, Küpkersweg 70, 26129 Oldenburg, Germany

**Correspondence:** Sonja Krüger (sonja.krueger@uni-oldenburg.de)

**Abstract.** The optimisation of the power output of wind turbines requires the consideration of various aspects including turbine design, wind farm layout and more. An improved understanding of the interaction of wind turbines with the atmospheric boundary layer is an essential prerequisite for such optimisations. Using numerical simulations, a variety of different situations and turbine designs can be compared and evaluated. For such a detailed analysis, the output of an extensive number of turbine and flow parameters is of great importance. Usually simulations are either specified to the output of turbine parameters or the detailed simulation of the flow. In this paper a coupling of the aeroelastic code FAST and the Large-Eddy Simulation tool PALM is presented. The advantage of the coupling of these models is that it enables the analysis of the turbine behaviour, i.a. turbine power, blade and tower loads, under different atmospheric conditions. The proposed coupling is tested with the generic NREL 5 MW turbine and the operational eno114 3.5 MW turbine. Simulating the NREL 5 MW turbine allows for a first evaluation of our PALM-FAST-coupling approach based on characteristics of the NREL turbine reported in the literature. The comparisons of the simulations to the NREL literature values show very promising results. Furthermore, a validation with free-field measurement data for the eno114 3.5 MW turbine for a site in Northern Germany is performed. The results show a good agreement with the free field measurement data. Additionally, our coupling offers an enormous reduction of the computing time, in comparison to similar methods with the same detail, and at the same time an extensive output of the turbine data.

## 1 Introduction

Wind energy poses a major contribution to today's renewable energy production (WindEurope, 2020). In this context, the prevailing atmospheric conditions, i.e. atmospheric stability with turbulence and shear, highly influence the power output of wind turbines and loads exerted on them (Doubrawa et al., 2019). Numerical simulations offer the possibility to study such effects in detail, but they are limited by the available computational capacity. However, the possibilities for numerical simulations in wind energy research are continuously expanded through the improvement of computational facilities, but also through the development of more efficient simulation tools.

With the help of Large-Eddy Simulations (LES) the influence of different stabilities (i.e. neutral, stable or unstable stratification) on the power production of wind turbines and the calculation of loads of a turbine can be investigated under controllable conditions, which is also the scope of the present work. A wide range of different stratified flows can be calculated with





LES, from stable, as shown in e.g. (Beare et al., 2006), (Kosović and Curry, 1998), to near-neutral (Porté-Agel et al., 2011), (Drobinski et al., 2007) to unstable (Maronga and Raasch, 2013). Differences depending on the atmospheric stability were investigated in several publications already (Dörenkämper et al., 2014) and (Wharton and Lundquist, 2012). In (Vollmer et al., 2016) a different development of the wake was observed depending on the atmospheric stability, which is an indication of a

dependence of the turbine behaviour also on atmospheric conditions. For LES different approaches for simulating the impact of wind turbines on the flow have been developed during the last decades. Furthermore, LES models can provide a variety of additional options like topography, an ocean model or tracing of particles, e.g. in (Maronga et al., 2015) and (Maronga et al., 2020). Depending on the LES model, it is therefore possible to simulate flow conditions in complex terrains or investigate the influence of the surrounding area on the turbine, c.f. footprint analyses (Steinfeld et al., 2008).

The comprehension gained from LES simulations also is a valuable basis to develop and validate less cost intensive models such as Reynolds-Averaged Navier Stokes (RANS) (Lübcke et al., 2001) or Kaimal-/Mann-models (Doubrawa et al., 2019). There are different ways to model a turbine, as can be seen in e.g. (Witha et al., 2014) and (Wu and Porté-Agel, 2013). They differ greatly in their level of detail and computing time requirements, one turbine model lacks turbine details, but is more time efficient and another model is more detailed, but has a higher demand on computational power. The models currently used to

calculate loads on entire wind turbines, like e.g. FAST (Jonkman and Buhl Jr., 2005) or Bladed (DNV GL, 2020), require wind fields as input, which are generally computed with comparatively simple models, like e.g. TurbSim (Jonkman, 2009a). TurbSim and comparable models commonly use the Mann-Model (Mann, 1998) or the Kaimal-Model, c.f. (Kaimal et al., 1972), (IEC, 2005), to model turbulence. These models assume Gaussian statistics and cannot display intermittency, which is found in real wind conditions and influences turbine loads, c.f. (Mücke et al., 2011).

Most commonly used turbine models are either an Actuator Line Model (ALM) or an Actuator Disc Model with rotation (ADMR) or without rotation (ADM). In an ALM the blades are simulated separately as lines in the flow, whereas in ADM and ADMR the rotor is modelled in the flow as a disk. In (Martínez-Tossas et al., 2015) a comparison between an ALM and an ADM showed a dependency of the results on the method of projecting the forces of the turbine into the flow and on the grid resolution. Further, an overestimation of the power output in the ALM at higher tip speed ratios was observed. As also shown

by (Churchfield et al., 2017), such an overestimation of power in turbine models can be caused by the projection of the forces and the sampling of the wind speed for calculating the turbine forces. In (Mittal et al., 2015) different methods of sampling the wind speed at the blade positions were tested and an influence on the power and thrust output was observed.

Simplifications can lead to a lack of information about either the atmospheric flow or the turbine behaviour and, thus, possibly less accurate results (Doubrawa et al., 2019). To address the problem of losing information of either the turbine or the flow

and provide a reliable tool, we present a newly developed computing routine here, with which it is possible to calculate LES in combination with a well resolved turbine model.

The objective of our work is to validate a further developed coupling method between the LES tool PALM (Maronga et al., 2015) and the aeroelastic model FAST (Jonkman and Buhl Jr., 2005), which is based on (Bromm et al., 2017), and to show how well the turbine behaviour in different atmospheric conditions is represented by this method. Such a coupling enables

detailed studies of turbine behaviour in complex situations while gaining extensive information about the turbine, like e.g.





turbine loads.

We make use of an Actuator Sector Method (ASM), where the blade movement is described as a segment of a circle. This allows for a larger time step in PALM than in FAST as the movement of the blade during that time step is captured in the area of the sector. A similar method is suggested in (Storey et al., 2015), where an ASM is tested in simulations. In (Storey et al., 2013) a coupling of FAST and an LES solver was described and investigated. In FAST itself an ALM is implemented, which in the case of (Storey et al., 2013) communicated with an ADM in the LES solver. (Storey et al., 2013) focused on the wake development, but not on the turbine parameters. In (Churchfield et al., 2012) a non-transient connection (meaning no continuous exchange of information) between an LES tool and the aeroelastic turbine model FAST was used for investigating the influence of wakes and atmospheric stability on turbine behaviour.

In this paper the enhanced coupling is presented. Furthermore, a systematic validation with measurement data for different atmospheric conditions with respect to a detailed set of variables is shown. A first comparison to other codes with a limited number of selected test cases, and without describing the coupling in detail, has been performed in the context of a joint study (Doubrawa et al., 2020).

In section 2 the enhanced coupling method is introduced and described, followed by simulations of the generic NREL 5 MW turbine in section 3.1 and the comparison to measurement data in section 3.2. The use of the generic NREL 5 MW turbine offers the opportunity to compare different models to each other with respect to the turbine output and computing times. To validate the proposed coupling and to assess the quality of the results, a non-generic turbine is simulated as well and compared to measurement data of a turbine situated in the north east of Germany.

With these comparisons, we show that the PALM-FAST coupling calculates realistic turbine output parameters, that this is not only valid for the global turbine parameters like power output, but also for individual component parameters like blade and tower loads and that the differences in the turbine behaviour due to different atmospheric conditions can be seen in the simulations as well. The final section 4 contains the conclusions and an outlook to subsequent work.

## 2    Methodology: The PALM-FAST coupling

In the present work, the aeroelastic turbine code FAST (Jonkman and Buhl Jr., 2005), developed at the National Renewable Energy Laboratory (NREL), USA, and the Large-Eddy simulation (LES) tool PALM (Maronga et al., 2015), developed at the Institute of Meteorology and Climatology (IMUK) of Leibniz University Hannover, are coupled. In addition to the power output FAST provides extensive information about the turbine response to the incoming flow, i.e. individual blade and tower loads, rotor speed, etc. PALM enables the simulation of an atmospheric flow for a wide range of different situations, like e.g. different stabilities using heating or cooling of the surface.

An earlier version of the coupling between FAST and PALM, described in (Bromm et al., 2017), was used here as a basis to be extended with respect to decreasing the computational time and improving the quality of the results. The previous implementation from (Bromm et al., 2017) was based on an ALM and required small time steps in both FAST and PALM. Also, it used the wind speeds at the rotor disk for calculation in FAST.





In an ALM the rotor blades are simulated as moving lines in the model domain and require a small computational time
step in order to calculate the movement andin order not to miss information at the fast moving blade tips. As the movement
of the blades is reproduced, an ALM can give information on the turbine in general, but also on separate blade data like
blade loads. A more computational time saving option is to simulate the turbine rotor as a disk, which is done in ADM
simulations. Additionally to the obstruction the rotor causes for the flow, a rotation can be added to the simulation (ADMR)
which increases the quality of the wake simulation. However, no information about individual blade parameters can be gained
in such a simulation.

To combine the advantages of both kinds of turbine models, i.e. the detailed output of the ALM and the low computational
costs of the ADMR, a so-called Actuator Sector Method (ASM) is used in this work.

PALM, when run in a normal set-up without FAST, uses either the Courant Friedrichs Levy (CFL) criteria or the diffusion
criteria to determine the largest possible time step, which in general is larger than a time step needed for a proper ALM
simulation. Therefore, using the same time step in both, FAST and PALM, affects the computational time required for the LES.
In the present work, we decouple the time step and allow the pure LES time step criteria (CFL and diffusion criteria), which
were mentioned above, to determine the time step in PALM and with this reduce the total computational time significantly.

In more detail, we use an ASM model for the projection of forces in PALM, whereas in FAST we still use the ALM model.
Through this set-up, the computing time can be reduced tremendously, since the more time consuming operations take place
in PALM and not in FAST. However, for simplicity, our whole coupling routine described in this work is simply abbreviated
as ASM hereafter.

Our ASM works as follows (see figure 1a): While FAST carries out small time steps $\Delta t_F$ as is necessary in an ALM, PALM
uses its own time step $\Delta t_P > \Delta t_F$ determined by the atmospheric model time step criteria. The simulation starts with FAST
communicating the initial blade positions. The wind speeds at these positions are determined from the wind fields simulated by
PALM and sent back to FAST. PALM then carries out one time step and is ahead in the simulation. Once PALM has calculated
its time step, the windfield is "frozen" and provides FAST with the wind speeds that are needed while FAST catches up and
calculates up to the current simulation time in PALM.

FAST therefore receives wind speeds of this frozen windfield and calculates the responding forces for the blades. During the
larger PALM time step, the rotor blades cover a segment of the rotor area, a sector. The width of the sector $\alpha$ is calculated
by the PALM time step $\Delta t_P$ and the rotor speed $\Omega$, which the FAST model communicates to PALM at the beginning of the
PALM time step, using $\alpha = \Omega \cdot \Delta t_P$. During the time step of PALM, several calculations of FAST are performed, similar to
the schematic in figure 1b. Except the values of the bold central line, the information of the forces at the positions of the
neighbouring lines are not used in PALM, but are output in FAST. The values of the bold central line are used for all of the $m$
lines in the sector, as in figure 1b. For each line a Gaussian shaped smearing is calculated and projected into the model domain.
This smearing of the forces is realised by a polynomial resulting in a Gaussian shape that distributes the forces over the area
surrounding the rotor blade in all three direction of space, c.f. (Sørensen et al., 1998):

$$\eta = \frac{1}{\varepsilon^3 \pi^{3/2}} \cdot \exp\left[-\left(\frac{r}{\varepsilon}\right)^2\right], \tag{1}$$





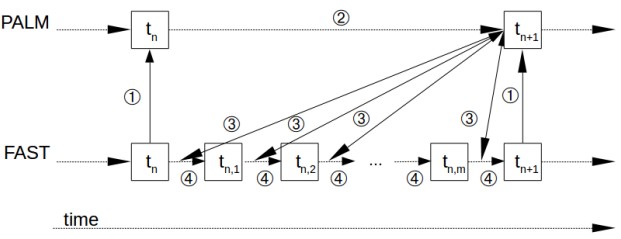

① Sending current positions and forces
② calculating flow field
③ Fast gives current blade position to PALM. PALM gives velocities at the blade positions to FAST.
④ Calculating the current turbine response (including new positions and forces).

(a) Schematic of the PALM and FAST time stepping.

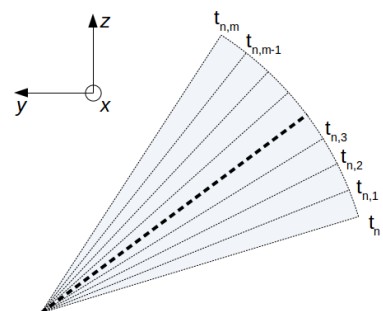

(b) Schematic of one circle segment of the ASM. The values of the bold central line are used for the projection of the forces into the flow. Here, $y$ and $z$ denote the rotor plane and $x$ the streamwise direction.

**Figure 1.** Schematic of the operation mode of the PALM-FAST ASM coupling.

where $\eta$ is the so-called regularisation function which is applied at the nodes of the grid within a certain vicinity of the turbine, $r$ is the distance between the respective node and the blade element from which the respective force stems and $\varepsilon$ is a factor of
the grid size that is typically set to $\varepsilon = 2 \cdot \Delta$ (Troldborg, 2008), with $\Delta$ being the grid size.

In general, the forces that occur at the blades are calculated based on the wind speed that is present at the blade position, i.e. the positions in the rotor plane. However, this wind speed does not represent the actual wind speed entirely as it depends on the grid resolution and has to be interpolated to the desired positions. Close to the last known blade positions this interpolation leads to errors and higher wind speeds than in reality, which leads to an overestimation of the power output. Additionally, the
projection width of the forces, i.e. the width defined by the regularisation function, influences the wind speed close to the blade immensely. To circumvent these issues, we take the wind speeds for the ASM in positions upstream of the turbine.

Far enough upstream of the rotor, the flow can be assumed to be almost undisturbed by the rotor. The wind speeds at the rotor area are then estimated using the induction model SWIRL of FAST. SWIRL uses the so-called Taylor's frozen turbulence hypothesis (Taylor, 1938) and calculates the induced velocity in axial and tangential direction. For further information see
(Moriarty and Hansen, 2005) and (Harman, 1994). In the current coupling a temporal change of the wind field as it approaches the rotor is not included. For the statistics of the turbine parameters this is not necessary, however, when the temporal sequence becomes relevant this can be resolved in the postprocessing of the results by shifting the results in time.

A comparison of different approaches, including the enhanced coupling described here, was done in (Doubrawa et al., 2020) to simulate site specific behaviour of a turbine. Besides LES, the discussed models also included Reynolds-Averaged Navier
Stokes (RANS) simulations and were compared with respect to turbine output and wake data in different atmospheric stabilities. The models performed differently depending on the simulation of the inflow conditions and the used resolution. Especially for the neutral case our coupling showed very good results.



## 3 Validation

The validation of the coupling is divided into two parts. The first part is the evaluation of results using the generic NREL 5 MW
turbine. The second part is the comparison to measurement data for a more extended analysis, for which a non-generic turbine
is simulated.

### 3.1 Validation of the coupling on the basis of the generic NREL 5 MW turbine

The NREL 5 MW turbine (Jonkman et al., 2009b) is a generic turbine which has been used extensively in simulations (Church-
field et al., 2012), (Storey et al., 2013), (Storey et al., 2015), (Vollmer et al., 2016), (Sathe et al., 2013), (Lee et al., 2012). The
155 NREL 5 MW turbine was developed by the National Renewable Energy Laboratory (NREL) and a FAST model of the turbine
is included in the FAST repository.

A comparison of four different methods is made. This includes a transient coupling between FAST and an ALM in PALM,
meaning the same time step size in FAST and PALM, (abbreviated as ALM). Furthermore, the ASM with two different modes
of retrieving the wind speed is used, namely the ASM with the described method of reading out wind speeds in front of the
160 turbine in combination with the induction model SWIRL (denoted as ASM), and also taking the wind speeds at the rotor area
without any induction model (denoted as ASM w/o SWIRL). As fourth method FAST on its own is used (denoted as FAST).
For FAST on its own, the inflow wind option "steady wind conditions" is used. To evaluate the different methods, at first, a
laminar case with the same wind speed over height is considered.

The LES simulations use a resolution of 5 m and $384 \times 192$ grid points in flow direction and perpendicular to flow direction,
respectively. In vertical direction, 192 grid points and a stretching are used, resulting in a total domain height of 3359 m. A
larger model domain of 384 grid points perpendicular to flow direction was tested as well to determine whether the size of
the model domain influences the results. However, no differences in the results were detected and therefore the smaller model
domain was used for the simulations. The boundary conditions at the in- and outflow are set to cyclic, however, only the time,
where the wake does not affect the inflow yet was evaluated. Additionally, the surface condition is set to a free slip condition.
The wind speed in the flow is set to $8 \, \mathrm{m \, s^{-1}}$. The inflow conditions for FAST are set accordingly.

In figure 2, a comparison of the generator power for the generic 5 MW NREL turbine is shown. The result calculated by FAST
coincides with the expected value, as published by NREL (Jonkman et al., 2009b). The ALM and ASM w/o SWIRL result in a
too high power output, which is assumed to be, most importantly, due to the wind speeds used to calculate the blade response
which is taken in the rotor plane. A further difference can be seen in the projection of the forces, which leads to different shapes
of the simulated rotor. As described above, in the rotor area there is the danger of reading out too large velocity values. The
ASM bypasses this issue by using the SWIRL induction method and results in a generator power which corresponds well with
the expected one. The ASM w/o SWIRL shows an even higher power output than the ALM. The reason for that may be that,
in the ASM w/o SWIRL the area that is blocked in the rotor area is larger than for ALM which might result in higher wind
speeds in between the sectors, like a nozzle. As the wind speeds next to the projected forces are used to calculate the turbine
response, these higher wind speeds would lead to a higher power output.





A turbulent case is calculated as well. However, no comparison to FAST alone is done here since there is no literature value available to compare the results with. For the turbulent case, a neutral flow is simulated with neither heating nor cooling of the surface. A resolution of $4\,\mathrm{m}$ is used with $1200 \times 480$ grid points in flow direction and perpendicular to flow direction, respectively. In vertical direction 192 grid points and a vertical stretching are used again, resulting in a vertical height of $1728\,\mathrm{m}$.

The roughness length is set to $0.05\,\mathrm{m}$, the wind speed at hub height is about $7.4\,\mathrm{m\,s^{-1}}$. In this simulation non-cyclic boundary conditions are used. If cyclic boundary conditions were used, the wake of the turbine would be fed into the inflow again and would, therefore, distort the flow in front of the turbine. In order to avoid this, PALM offers the opportunity of non-cyclic boundary conditions and a turbulence recycling method, for more information see (Maronga et al., 2015).

Figure 3 shows the time series of the generator power. The wind speeds in the ASM are taken $2D$ in front of the turbine, which

in this example is a distance of $252\,\mathrm{m}$, resulting in a time shift of the flow reaching the turbine of about $34\,\mathrm{s}$. Therefore, when comparing the turbine output the result of the ASM simulation is shifted by $34\,\mathrm{s}$ for a better comparison to the other results. As for the laminar case, the ASM leads to a lower power output than the other models, whereas the differences are comparable to the laminar case in 2. Also, roughly the same peaks and therefore structures of the flow are present in the ASM results. This implies, that the coupling works in a turbulent environment as well.

Furthermore, these simulations are used to compare the computational times of the ALM and ASM. In the laminar case the ASM is nine times faster than the ALM while using the same amount of cores, i.e. the computational time is reduced by up to 89%. The turbulent case is calculated with a difference in the allocated cores: the ALM uses four times more cores than the ASM, however the ASM is still about 3.5 times faster than the ALM. Consequently, the ASM provides the same set of output parameters as the ALM, but is significantly faster.

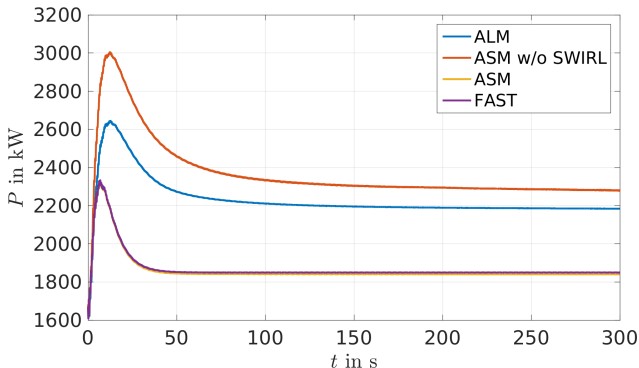

**Figure 2.** Comparison of different simulation methods for the generator power of the $5\,\mathrm{MW}$ NREL turbine in a laminar flow with $8\,\mathrm{m\,s^{-1}}$ wind speed.





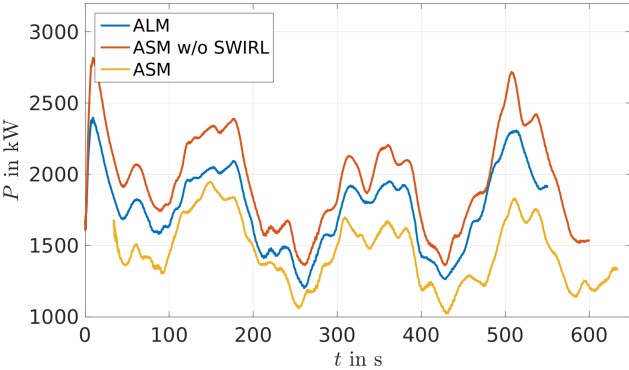

**Figure 3.** Comparison of different simulation methods for the generator power of the 5 MW NREL turbine in a turbulent flow at about $7.4\,\mathrm{m\,s^{-1}}$ wind speed at hub height.

## 3.2 Validation of the coupling with the eno114 3.5 MW turbine

As the generic NREL 5 MW does not allow for a comparison to measurement data, a free-field turbine is used for further analyses. Measurement data of an eno114 3.5 MW turbine with a hub height of 92 m and the corresponding FAST turbine model are used for further investigations.

First, we consider laminar cases with uniform wind speed over height for the eno114 3.5 MW in order to establish a power curve. The reference power curve is obtained from stand-alone FAST runs, with a laminar inflow. The FAST turbine model is provided by eno. The calculated reference power curve coincides well with the published power curve of eno (eno energy, 2019). For a wind speed of $8\,\mathrm{m\,s^{-1}}$ the different models are compared again (c.f. figure 4). The ALM again shows a higher power output than the reference power curve, the ASM coincides with the reference value and therefore with the value published by eno. The ASM w/o SWIRL shows again a higher power output than the ALM, although the difference is not as significant as in the laminar case of the NREL 5 MW turbine (c.f. figure 2).

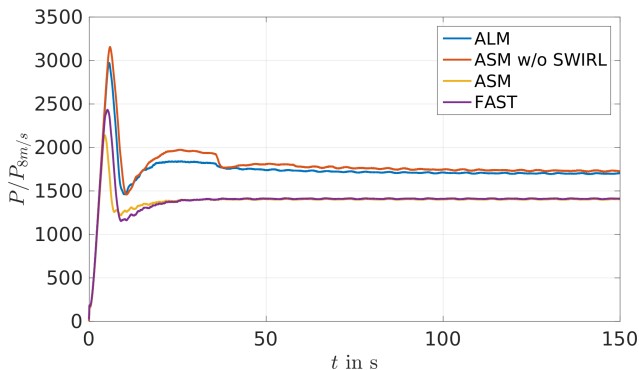

**Figure 4.** Comparison of different simulation methods for the generator power of the eno114 3.5 MW turbine in a laminar case with a wind speed of $8\,\mathrm{m\,s^{-1}}$, normalised by the respective value of the eno power curve at $8\,\mathrm{m\,s^{-1}}$.





### 3.2.1 Conditions at the onshore measurement site near Brusow

The onshore measurement site, from which data was available, is situated in Northern Germany close to the village of Brusow. At the measurement site two eno114 3.5 MW turbines are present. For one turbine (turbine 1 in figure 5) measurement data was available, consisting i.a. of the power output, rotor speed, generator speed and tower, main shaft and blade root bending moments.

Apart from the two eno turbines the measurement site was also equipped with a met mast. Figure 5 shows the general set-up of the site. The met mast contained three cup anemometers, one wind vane and one eddy-covariance stations of type IRGASON from Campbell Scientific. The cup anemometers were situated at the heights 34.6 m, 89.3 m and 91.5 m, the wind vane at 89.3 m and one of the eddy-covariance stations at 34.6 m. Another eddy-covariance station was located at a height of 2.3 m on the boom of a separate tripod that was situated next to the met mast.

From the 20 Hz data provided by the eddy-covariance stations, turbulence statistics with a resolution of 30 minutes are obtained by applying the eddy-covariance Software TK3 (Mauder and Foken, 2015). The planar fit method (Wilczak et al., 2001) is used for correcting impacts of a tilted device on the turbulence statistics. For calculating the planar coefficients the whole available data set is taken into account. As the IRGASON is not an omni-directional device, planar fit coefficients are calculated for four different wind direction sectors as suggested by the manufacturer of the IRGASON. The distance of the met mast to the turbine, for which measurement data is available, was 280 m ($\approx 2.5D$) in direction 190° referring to the wind turbine.

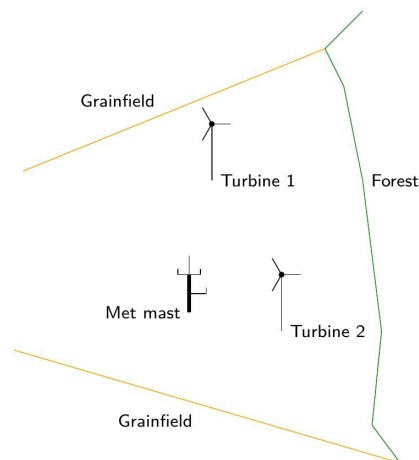

**Figure 5.** Schematic of the measurement site in Brusow.

Data of all sensors is available from 10. May until 30. June in 2017. To the east of the site of the turbines and met mast a forest is located which influences the measurements greatly. Therefore, the measurement data is filtered for the westerly wind directions, where mostly grainfields are situated.



We estimate the roughness length of the surrounding area using the wind speed $u_{ec}$ and the friction velocity $u_*$, both provided

by the lower eddy-covariance station, with equation 2 for data of neutral stratification, where $k$ is the von Kármán constant, $z_{ec}$ the height of the respective eddy-covariance station and $z_0$ the desired roughness length:

$$u_{ec} = \frac{u_*}{k} \ln \left( \frac{z_{ec}}{z_0} \right). \tag{2}$$

The plot of the roughness length distribution (figure 6) shows an approximate roughness length of $z_0 = 0.1\,\mathrm{m}$ for the westerly region. This value corresponds to farmland and hedges in the summer time according to Stull (2003), which is in agreement with the plants on site and therefore $z_0 = 0.1\,\mathrm{m}$ is a reasonable value for the roughness length. From the data of the eddy-covariance

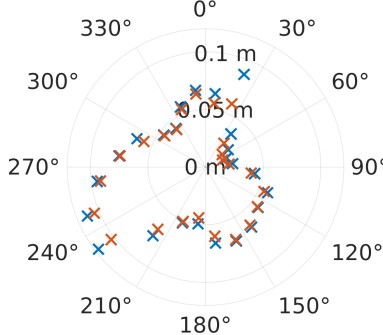

**Figure 6.** Roughness length distribution for varying wind directions for the measurement period. Two methods of averaging the roughness length values gained by equation 2 were used, here $z_0$ denotes the roughness length determined from 30 min eddy-covariance data and $j$ denotes the 15° wind direction bins: 1. averaging $z_{0,j}$ per 15° bins (blue) and 2. averaging using $\ln z_{0,j} = \frac{\langle u_* \ln(z_{0,j}) \rangle}{\langle u_* \rangle}$ per 15° bins (red). The $\langle ... \rangle$-brackets denote the average over values within the 15° bins.

stations the stability parameter $\frac{z}{L}$, with $z$ as the measurement height, here $z_{ec}$, and $L$ the Obukhov length, are obtained from the application of the software TK3 to it. In the following, the power that was produced during the respective times is plotted with respect to the wind speeds filtered by the stability, calculated from the data collected by the eddy-covariance stations. For that, the 50 Hz measurement data of the power is averaged over 10 min intervals, denoted as $P_{10}$. These 10 min power values are

sorted according to stability and wind speed and averaged according to the wind speed within the respective stability, resulting in $\bar{P}_{10}$. For normalisation the maximum 10 min power value $P_{10\,max}$ is used. Accordingly, the standard deviation is calculated, i.e. the standard deviation is calculated for 10 min intervals $\sigma_{P10}$, then these 10 min values are averaged according to their stability and wind speed $\bar{\sigma}_{P10}$ and normalised with the maximum 10 min standard deviation value $\sigma_{P10\,max}$.

Figure 7 shows the resulting power curve, which is analysed with respect to the stability. Due to the relatively low number

of measurements, the stabilities, based on the data of the lower eddy-covariance station of $z_{ec} = 2.3\,\mathrm{m}$, are sorted for stable $(\frac{z}{L} > 0.0115)$, neutral $(-0.0115 < \frac{z}{L} < 0.0115)$ and unstable $(\frac{z}{L} < -0.0115)$, but not for further classification in very stable and very unstable , c.f. table 1.

It can be seen that the measurement data coincides very well with the power curve provided by eno for the 3.5 MW turbine (c.f. (eno energy, 2019)). Also, no significant differences between the different stratifications can be observed. However, differences





for the stratifications can be seen in the turbulence intensity and the shear (c.f. figures 9 and 10). As expected the unstable cases have a higher turbulence intensity (TI) than the stable cases. This is also visible in the standard deviation of the power (c.f. figure 8), as the higher TI in the neutral and unstable case leads to a higher standard deviation of the power than in the stable situations with lower TI.

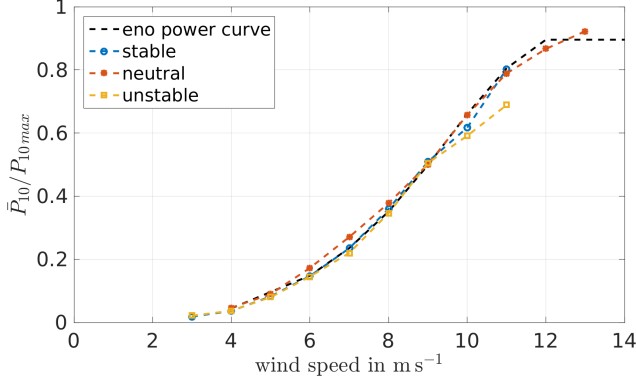

**Figure 7.** Power curves determined from the measurement data for May/June 2017, normalised by the maximum 10 min power, for different stabilities (determined from eddy-covariance data) and plotted in comparison to the eno power curve determined by FAST in laminar conditions.

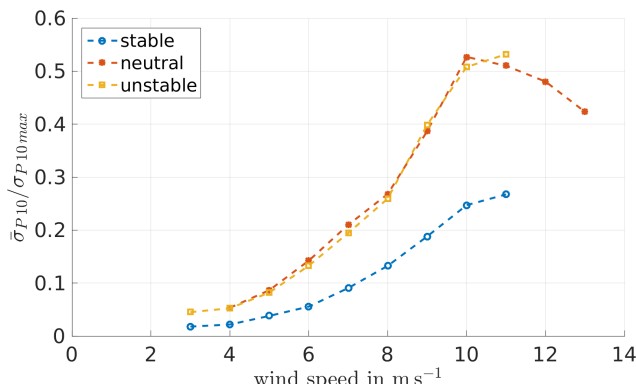

**Figure 8.** Standard deviation for 10 min intervals of the measured turbine power output, calculated according to: $\sigma_{P10} = \sqrt{\frac{1}{N_{meas}-1} \sum_{k=1}^{N_{meas}} |P(t_k) - P_{10}|^2}$, where $P(t_k)$ denotes the power data measured in 50 Hz, $P_{10}$ the 10 min average and $N_{meas}$ the number of measurements within the 10 min interval, normalised by the maximum 10 min standard deviation of the power, for May/June 2017, sorted and averaged according to stability (determined from eddy-covariance data) and wind speed.





**Table 1.** Classification of atmospheric stability according to Obukhov length $L$, based on (Peña et al., 2008).
The distribution of the atmospheric stability in the measured data can be seen in figures 9 and 10.

| Obukhov length [m] | Atmospheric stability |
|:---:|:---:|
| $10 \leq L \leq 200$ | Stable |
| $|L| \geq 200$ | Neutral |
| $-200 \leq L \leq -50$ | Unstable |

### 3.2.2 Simulation set-up for Brusow

In the following, the simulation set-ups for PALM and FAST that are used for the comparison to the measurement data are described.

**PALM**

In order to compare simulation results to the measurement data, simulations are computed that result in flow conditions similar to those observed under neutral boundary layer (NBL) and stable boundary layer (SBL) flow at Brusow. As can be seen in 260 figures 9 and 10 most data is available for the NBL and slightly SBL.

Precursor simulations without a turbine are performed in order to reach a stationary state and evaluate the produced inflow conditions prior to the main simulations containing a wind turbine. The resolution for both, neutral and stable conditions, is set to 4 m in $x$- and $y$-direction and in vertical direction up to a height of 600 m. Above $z = 600$ m a vertical stretching of the grid with a factor of 1.08 is used. In accordance with the results of our evaluation of the roughness length from eddy-covariance 265 data at the site, the roughness length $z_0$ is set to 0.1 m, (c.f. figure 6). A homogeneous roughness length is set in the model domain and no topography is taken into account, which means that idealised simulation conditions are used. In table 2 the different set-ups and in table 3 the resulting flow conditions are shown.

**Table 2.** Setup of the precursor simulations: Size of the model domain in streamwise $x$, spanwise $y$ and vertical $z$ direction, grid size $\Delta$, cooling rate $\Delta\Theta/\Delta t$, geostrophic wind speed components at the surface in $x$- and $y$-direction $u_g$, $v_g$ and total simulated time $t_{end}$.

| | $x$ | $y$ | $z$ | $\Delta$ | $\Delta\Theta/\Delta t$ | $u_g$ | $v_g$ | $t_{end}$ |
| | [m] | [m] | [m] | [m] | [K h$^{-1}$] | [m s$^{-1}$] | [m s$^{-1}$] | [s] |
|:---:|:---:|:---:|:---:|:---:|:---:|:---:|:---:|:---:|
| NBL | 5184 | 2304 | 2928 | 4 | 0 | 10.0 | -4.25 | 93600 |
| SBL | 1440 | 960 | 616 | 4 | -0.25 | 9.5 | -5.17 | 46800 |





**Table 3.** Resulting flow parameters after reaching a stationary state in the precursor simulations, averaged over 3600 s: The magnitude of the wind speed at hub height averaged over the model domain $U_{92m}$, turbulence intensity calculated at one position in 92 m height $TI_{92m}$, shear parameter $\alpha$ (based on the power law $u_2 = u_1 \left(\frac{z_2}{z_1}\right)^\alpha$ for the relation of wind speeds at different heights), Obukhov length $L$ in a height of 2.3 m and boundary layer height $z_i$.

|  | $U_{92m}$ | $TI_{92m}$ | $\alpha$ | $L$ | $z_i$ |
|---|---|---|---|---|---|
|  | [m s$^{-1}$] | [%] | [ ] | [m] | [m] |
| NBL | 8.6 | 10.1 | 0.15 | 1228698 | 550 |
| SBL | 8.4 | 5.6 | 0.28 | 102 | 180 |

For the respective main runs including the turbine a larger model domain and non-cyclic boundary conditions were used to avoid influences of the wake onto the turbine. The model domain of the neutral case is larger than the one of the stable case, as in neutral conditions the turbulent structures tend to be larger than in stable conditions: The neutral model domain is set to 7680 m × 2595 m × 2928 m, the stable is set to 5760 m × 2304 m × 616 m. The simulations are set up according to the simulations in (Vollmer et al., 2016).

To reduce local effects caused by possible persistent structures in the flow, the main run is simulated three times with three different turbine positions in $y$ direction, respectively. Table 4 shows the differences of the flow between the turbine positions. The power output resulting from the simulations at the different positions are used to be compared to the measured data, yielding three results for both stabilities, respectively, as can be seen in figures 11 to 16.

**Table 4.** Turbine positions along the $y$-axis (keeping the same $x$ position), with the $y$-direction spanning from 0 m to 2595 m for the NBL and from 0 m to 2304 m for the SBL, in the model domain of the main run, additionally, the local wind speed $U_{92m}$ and turbulence intensity $TI_{92m}$ at hub height at these $y$-coordinates, taken $2.5D$ in front of the turbine averaged over the last 10 min of a 650 s simulation.

|  | $y$ | $U_{92m}$ | $TI_{92m}$ |
|---|---|---|---|
|  | [m] | [m s$^{-1}$] | [%] |
| NBL | 500 | 8.21 | 10.3 |
|  | 1000 | 8.92 | 10.5 |
|  | 1700 | 8.87 | 8.0 |
| SBL | 1000 | 8.32 | 6.0 |
|  | 1200 | 8.22 | 5.6 |
|  | 1600 | 8.23 | 6.2 |

Figures 9 and 10 show how the precursor simulations, i.e. the inflow conditions for the turbine, compare to the measurement data. The crosses represent the data from the precursor runs, so the undisturbed inflow averaged over space, at height 92 m, and time. In comparison to the measurement data, both simulations, neutral and stable, are in the lower region of the measured





turbulence and shear. However, the TI of the simulations is calculated using the resolved turbulence and disregarding the subgrid scale one, hence it is likely that the TI in the simulations is slightly higher than seen here. Therefore, the simulation set-up seems to resemble the inflow conditions at Brusow reasonably well. Since the flow conditions in the simulations match the measurements, the turbine output is compared in the following.

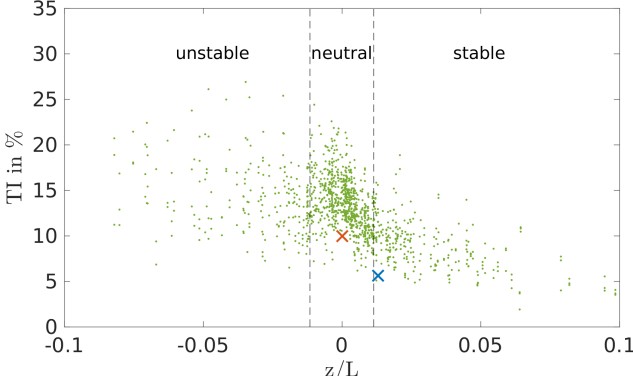

**Figure 9.** Turbulence intensity $TI_{92m}$ of the measurement data (green) in comparison to the resulting $TI_{92m}$ of the precursor runs sorted in neutral and stable (red - neutral, blue - stable).

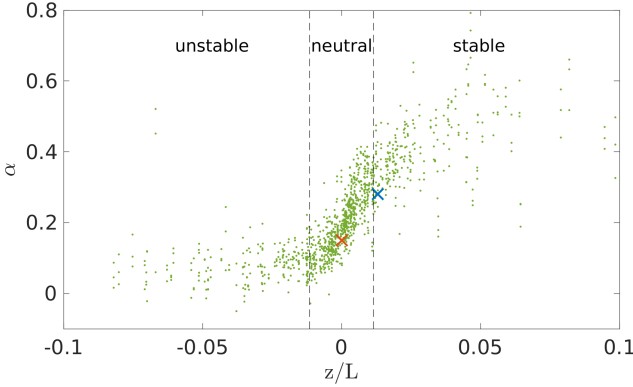

**Figure 10.** Shear of the measurement data (green) in comparison to the resulting shear of the precursor runs sorted in neutral and stable (red - neutral, blue - stable).

### FAST

The turbine model of the eno114 3.5 MW turbine for FAST was provided by eno, including structural information and a pitch, a speed and a yaw control in the format of a Bladed .dll file. However, the yaw of the turbine is neglected, as the flow in PALM was directed in such a way that the turbine is aligned with the wind. In FAST the modules ElastoDyn, AeroDyn and ServoDyn were used, the degrees of freedom for the blade and tower were set to true except the rotor-teeter and yaw flag. All the platform degrees of freedom were neglected, i.e. set to false. The time step throughout all modules was set to $\Delta t = 0.01\,\mathrm{s}$.



### 3.2.3  Comparison of the turbine data

In the following plots the output data of the turbine in the simulations is compared to the measurement data. The main runs of the simulations are run for a simulation time of $650\,s$ respectively, the results are averaged over $600\,s$, discarding the first $50\,s$ as a spin-up of the turbine simulation, this time frame is derived from the laminar case, c.f. figure 4. To compare the power output of the simulations to the measurement data, the power needs to be set into relation with the correct corresponding inflow wind speed. As the wind speed in Brusow is determined from a cup anemometer on a met mast in a distance of $2.5D$ from the turbine at hub height, in the simulation the wind speed is taken as well in a single point in a distance of $2.5D$ in front of the turbine position at hub height and averaged over time.

In figure 11 the simulation results are shown in comparison to the power curve determined by the measurement data at hub height. The error bars show the standard deviation of $10\,min$ means. Figure 12 shows the same plot enlarged at the wind speeds of the simulation. According to (Dörenkämper et al., 2014), using offshore data, and (Wharton and Lundquist, 2012), using onshore data, slight differences of the power output depending on the atmospheric stabilities can be seen. However, both publications together do not show a clear trend which stability generally leads to the higher power output. In an offshore environment, as in (Dörenkämper et al., 2014), unstable conditions lead to a higher power output below rated wind speed and at an onshore site, c.f. (Wharton and Lundquist, 2012), the stably stratified atmospheric boundary layer (ABL) yields the higher power output. However, different wind speeds were used as a reference, which makes a comparison of the results difficult. In (Wharton and Lundquist, 2012) a rotor equivalent wind speed was used, while (Dörenkämper et al., 2014) used the measurement data of a met mast at $90\,m$ height.

The measurement data of Brusow, with the wind speed at hub height as reference, does not show any clear tendency for the dependency of the wind turbine power on atmospheric stability, c.f. figures 7 and 11. A power curve depending on the rotor equivalent wind speed was calculated from the measured data as well, but does not conclude in a clear trend either. The rotor equivalent wind speed was computed according to (Wagner et al., 2014), but due to the limited number of measurement heights and their irregular distribution over the height, the results could be prone to errors. Therefore, for further analysis the hub height wind speed is used. The apparent independence of the wind turbine power on atmospheric stability might be due to the limited amount of only two months of data that was available or might be depending on the measuring and classification of the stability. As shown in (Wharton and Lundquist, 2012) the stability filtered power curve greatly depends on the measurement heights used for determining the shear. However, this behaviour is also not present in the simulations. Therefore, in our case, the power output is not the proper parameter to show different turbine responses depending on the atmospheric stability.

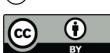

WIND
ENERGY
SCIENCE
DISCUSSIONS

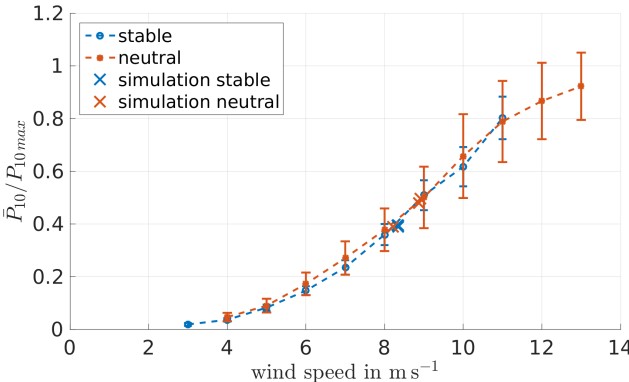

**Figure 11.** Power curve, normalised by maximum 10 min power, determined from measurement data including standard deviation in comparison to the results of the simulation (marked by ×). The standard deviation is plotted again in figure 13.

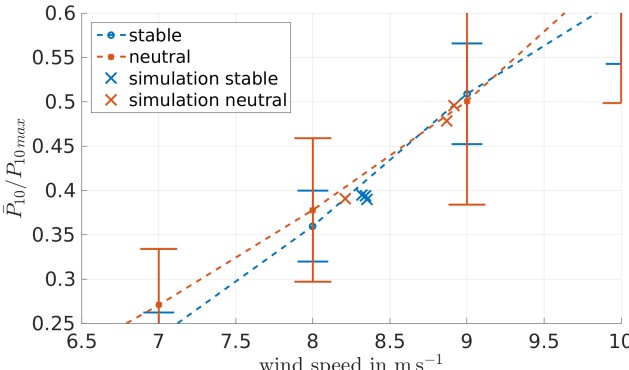

**Figure 12.** Enlargement of the normalised power curve determined from measurement data including standard deviation in comparison to the results of the simulation (marked by ×).

Figure 13 shows the standard deviation of the power with respect to the wind speed. Higher fluctuations of the power in the neutral cases can be observed, corresponding to the higher TI that is present in the neutral stratification, c.f. (Mittelmeier et al.,
2017). The simulation data shows a comparable behaviour with lower fluctuating power in the stable cases than in the neutral ones. In the three neutral simulations the distribution of the standard deviation is spread relatively wide compared to the stable cases. The three different positions that were used for the neutral simulations differ slightly in wind speed and TI, which is not the case for the stable cases (c.f. table 4).

To check whether this distribution is comparable to the measurement data, a plot of the standard deviation of the power with
respect to the TI is made (figure 14). It shows the relation between the power fluctuations to the TI for all measured values (green dots) and specifically the measured stable and neutral cases (blue and red asterisks) and in comparison the respective values of the simulations (blue and red crosses). The results of the simulations correspond well with the measurement data, therefore other turbine parameters available are compared. In specific, the blade and tower loads are investigated below.





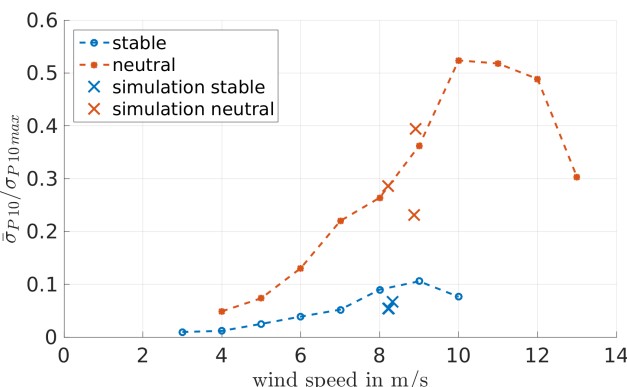

**Figure 13.** Normalised standard deviation of the power with respect to the wind speed determined from measurement data in comparison to the simulation results (×). Sorted into stability by eddy-covariance data, TI and shear determined from the met mast data.

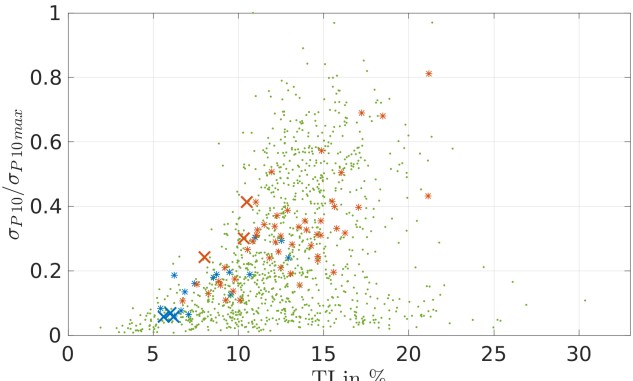

**Figure 14.** Standard deviation of the power with respect to the TI determined from measurement data (green – all wind speeds, blue and red asterisks – stable and neutral measurements at wind speeds of 8-9 m s$^{-1}$) in comparison to the simulation results (×).

The flap- and edgewise blade root bending moments respectively are evaluated, but also data for the tower top and base loads
is available and examined. Figures 15 and 16 show the measured blade root bending moments with respect to the wind speed,
the results of the simulations are indicated by crosses. The out-of-plane blade root bending shows a good agreement, the in-plane blade root bending moment differs a bit more. However, a more suitable way to compare the loads is to look at the spectra.



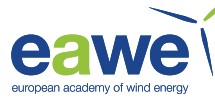 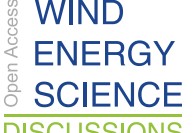

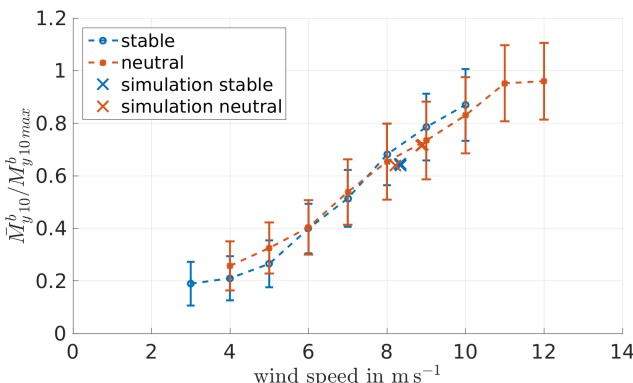

**Figure 15.** Blade root bending out-of-plane moment $M_y^b$ with respect to wind speed in comparison to the simulation results, averaged 10 min values sorted into stability and averaged according to wind speed, normalised with the maximum measured moment.

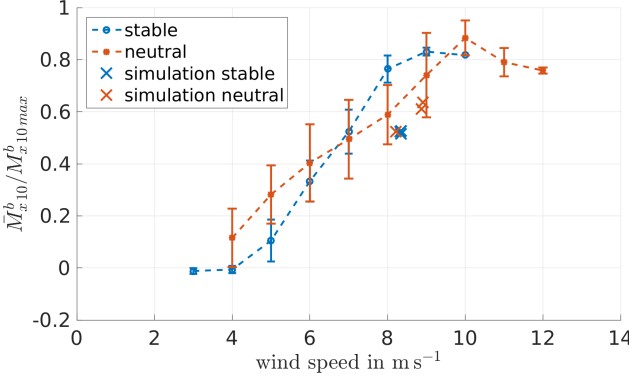

**Figure 16.** Blade root bending in-plane moment $M_x^b$ with respect to wind speed in comparison to the simulation results, averaged 10 min values sorted into stability and averaged according to wind speed, normalised with the maximum measured moment.

We filtered the data with respect to westerly winds, stability and rotor speed. The analysis of the rotor speed showed a difference

in the controller behaviour of the real system to the modelled one. This can be seen in figures 26 and 27. Apparently, the rotor speed curve at the start of the peak shaver region is slightly different (c.f. figure 27). Therefore, it is only possible to compare loads at either the same rotor speed or the same wind speed.

For the stable case some of the time intervals have to be discarded due to a varying quality of the load sensors, leaving one interval for the stable case where data is continuous for the blade and tower moments. For the neutral case the longest remaining

340 interval covers a span of 165 s long. The conditions of the chosen intervals are shown in table 5. Ideally the chosen intervals should match the simulation parameters, but due to the described limitations in the measurement data, the remaining intervals can be seen as the best fit. These available cases suffice for the validation of our code. For an even more detailed load analysis, better fits might be necessary.





**Table 5.** Summary of the parameters of the measurement interval data used for the spectra of the blade and turbine loads: wind speed at hub height $U_{92m}$, turbulence intensity at hub height $TI_{92m}$, shear parameter $\alpha$ and the length of the available time interval $t_{interval}$.

|  | $U_{92m}$ | $TI_{92m}$ | $\alpha$ | $t_{interval}$ |
|---|---|---|---|---|
|  | $[\text{m s}^{-1}]$ | [%] | [ ] | [s] |
| stable | 7.7 | 9.0 | 0.27 | 600 |
| neutral | 9.4 | 15.5 | 0.27 | 165 |

In the following the stable case will be discussed in detail. The neutral case also shows a good agreement between simulation and measurement data, but covers only a short time interval of only 165 s, the corresponding spectra can be found in the Appendix A.

Figures 17 and 18 show spectra of the blade root bending moments for the stable case. Figures 19 to 23 show the resulting tower loads spectra for the stable case.

The spectra are normalised by their maximum value: the blade root bending moments are normalised by the same value and tower top and tower base bending moments respectively with their respective maximum values as well. The frequency is normalised by the rotor speed $\Omega$.

In the spectra of the stable case it can be observed that the torsion loads show comparable results, c.f. figure 23. Also the fore-aft and side-to-side tower loads, c.f. 19 to 22, and the blade root bending moments, c.f. figures 18 and 17, are represented well in the simulation. In general, most of the multiples of the rotor speed are represented in both the measurements and the simulations and also their levels are comparable. The peaks show a difference in the width depending on the turbulence intensity, i.e. in the stable, less turbulent case the peaks are less wide than in the more turbulent, neutral case (figures in Appendix A). This can be observed both in the measurement data and the simulation results.

It can also be seen, that the 1P peak is of different height in the tower load spectra. The peak of the simulation data reaches higher, than the one of the measurement data. This is probably due to an overestimated blade imbalance in the simulation which has been used to respect weight and pitch differences between the blades, c.f. (Zhang et al., 2015). In the FAST turbine model one of the blades has a 1% higher mass density than the others and also a pitch offset of $0.3°$ is set between all three blades. This results in a very pronounced 1P peak that is not existing in the measurement data.

Notable is also that there seems to be a discrepancy between the simulation and measurement data in the tower top side-to-side bending moment in stable and neutral conditions. This might be caused by the difference in the tower model to the real behaviour of the turbine tower. It can be seen that the first tower eigenfrequency is slightly lower on the real turbine and therefore more prone to the rotational excitation. In the measurement data the first tower eigenfrequency is closer to the 1P peak and therefore the vibrations less damped.

Differences can also be observed in the 6P peak, especially in figures 19 and 20. The 6P peak is greatly influenced by the shear and the wind speed differences across the rotor area. A plot of the wind speed profiles can be found in the Appendix B, even though the shear is similar, the difference in the wind speeds, which are caused by the above described limitations in the





measurement data, led to diverging wind speed profiles.

In figures 24 and 25 a comparison between the neutral and stable simulation results for a blade root bending and tower bending moment, respectively, is shown. The bending moments that are mostly affected directly by the flow, i.e. by the thrust, are chosen. It can be observed that the neutral simulation leads to wider peaks due to the higher TI and the resulting varying rotor

speed. Also, a difference in the height and depth of some peaks can be seen. Namely for the blade root bending out-of-plane moment the 2P and for the tower fore-aft bending moments the 3P and 6P peaks are higher and reach further down for the stable case than the neutral case. These multiples of the rotor speed are influenced by the shear of the flow which also indicates a difference in the inflow of the turbines.

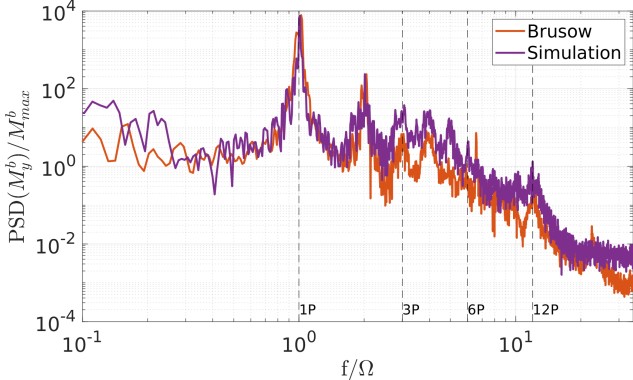

**Figure 17.** Spectrum of the blade root bending out-of-plane moment $M_y^b$ in comparison to the simulation results (stable). The data is normalised by the maximum value of the blade root bending moments.

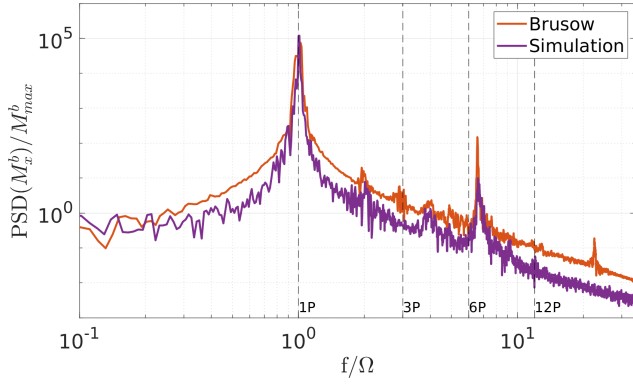

**Figure 18.** Spectrum of the blade root bending in-plane moment $M_x^b$ in comparison to the simulation results (stable). The data is normalised by the maximum value of the blade root bending moments.





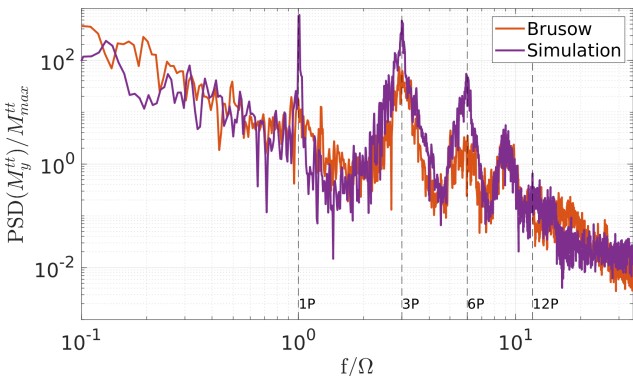

**Figure 19.** Spectrum of the tower top bending moment in fore-to-aft direction $M_y^{tt}$: Comparison of the measurement data to the simulation results (stable). The data is normalised by the maximum value of the tower top bending moments and the frequency is normalised by the rotor speed.

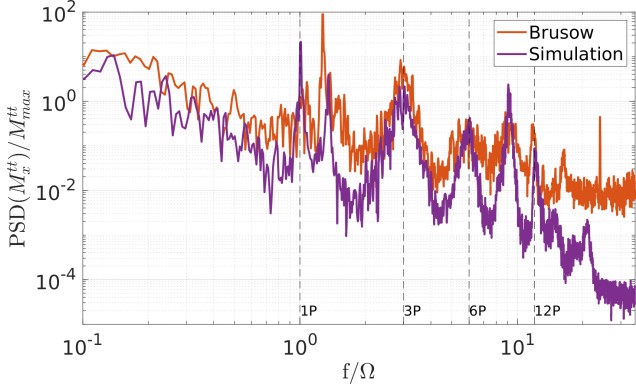

**Figure 20.** Spectrum of the tower top bending moment in side-to-side direction $M_x^{tt}$: Comparison of the measurement data to the simulation results (stable). The data is normalised by the maximum value of the tower top bending moments and the frequency is normalised by the rotor speed.

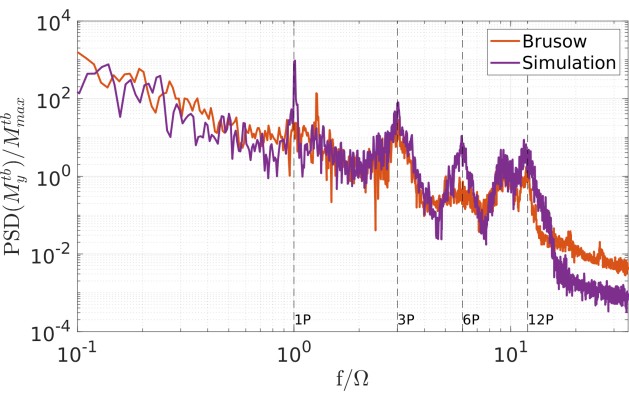

**Figure 21.** Spectrum of the tower base bending moment in fore-to-aft direction $M_y^{tb}$: Comparison of the measurement data to the simulation results (stable). The data is normalised by the maximum value of the tower base bending moments and the frequency is normalised by the rotor speed.

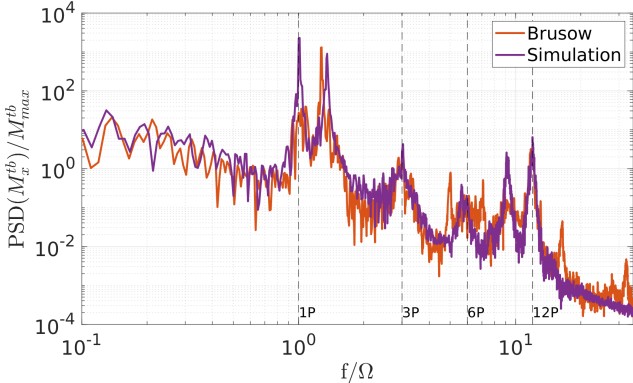

**Figure 22.** Spectrum of the tower base bending moment in side-to-side direction $M_x^{tb}$: Comparison of the measurement data to the simulation results (stable). The data is normalised by the maximum value of the tower base bending moments and the frequency is normalised by the rotor speed.

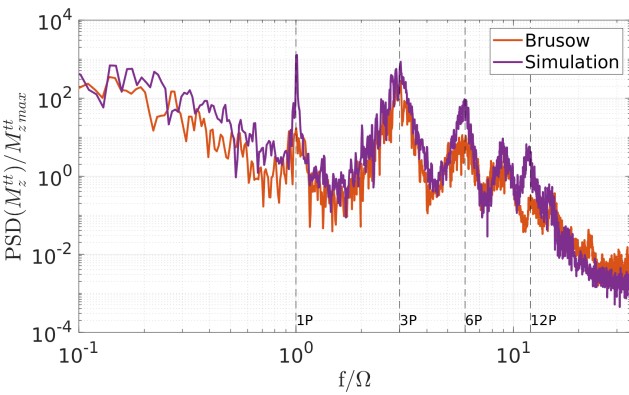

**Figure 23.** Spectrum of the tower top torsion moment $M_z^{tt}$: Comparison of the measurement data to the simulation results (stable). The data is normalised by the maximum value of the tower top torsion moment and the frequency is normalised by the rotor speed.





To investigate the loads further, rainflow counts and the value of the equivalent load range $\Delta_{eq}$ (non-normalised damaged
equivalent loads (DEL)) were calculated. Equation 3 shows the used Palmgren-Miner rule, taken from (Vera-Tudela and Kühn,
2017):

$$
\Delta_{eq} = \left( \sum_{k=1}^{n} N_k \Delta S_k^m / N_{ref} \right)^{1/m},
\tag{3}
$$

where $n$ is the number of different loading amplitudes, $N$ the number of cycles and $\Delta S$ the loading amplitude. Further, a
Wöhler exponent of $m = 10$ for the blades, $m = 4$ for the tower and a reference number of cycles $N_{ref} = 10^7$ is assumed.

A comparison between the measurement data and the simulation results is not useful in this case as the available intervals
vary in their inflow parameters and therefore the rotor speed. However, a comparison between the results of the simulation
of the neutral and stable boundary layer flow, respectively, show the influence of the stability on the load outputs of the LES
coupling. Table 6 shows the comparison of the equivalent load range for the stable and neutral simulations, calculated for a
10 min interval respectively. It can be observed that almost all the neutral values are higher than the ones from the simulation
of the stable case. The only exception is the blade root bending in-plane load, which shows approximately the same value for
both cases. As this load is not that dependent on the flow, but rather influenced by gravity and rotor speed, the result still seems
conclusive.

The values can be linked to the power spectra shown in figures 24 and 25. Particularly in the range of the lower frequencies
larger PSD values are obtained for the neutral case in comparison with the stable case. To investigate the influence of the lower
frequencies on the equivalent load range, the equivalent load range for the tower top fore-aft bending moment is calculated
with a high pass filter as an example. The following values result for the equivalent load range when the frequencies below 0.1
are disregarded: stable: $\Delta_{eq} = 81.8 \cdot 10^5 \, \text{kNm}$ and neutral: $\Delta_{eq} = 98.7 \cdot 10^5 \, \text{kNm}$, which clearly shows that the lower frequency
range has a great influence on the equivalent load range. A higher value for the neutral case is expected as the flow contains
larger eddies than the stable case.

This should be considered as a qualitative result. For a final quantitative analysis simulations with considerably larger run times
or a number of simulations with different seeding would be required. Also, in the papers (Lee et al., 2012) and (Holtslag et al.,
2016) no clear results are visible, in (Lee et al., 2012) it is stated that mainly the roughness has an influence on the loads, while
the stability has only a small effect. In (Holtslag et al., 2016), on the other hand, a clear influence of stability on the loads is
observed.





**Table 6.** Comparison of the equivalent load range $\Delta_{eq}$ of the simulation results (10 min interval), according to equation 3 with $m = 10$ for blade loads and $m = 4$ for tower loads.

| Load | $\Delta_{eq}$ [kNm] | |
|---|---|---|
| | stable | neutral |
| Blade root bending in-plane | 1579 | 1578 |
| Blade root bending out-of-plane | 687 | 717 |
| | $\Delta_{eq}/10^5$ [kNm] | |
| Tower top fore-aft bending | 70.6 | 132.7 |
| Tower top side-to-side bending | 1.1 | 8.9 |
| Tower top torsion | 82.0 | 154.3 |
| Tower base fore-aft bending | 910.0 | 7623.8 |
| Tower base side-to-side bending | 373.6 | 963.7 |

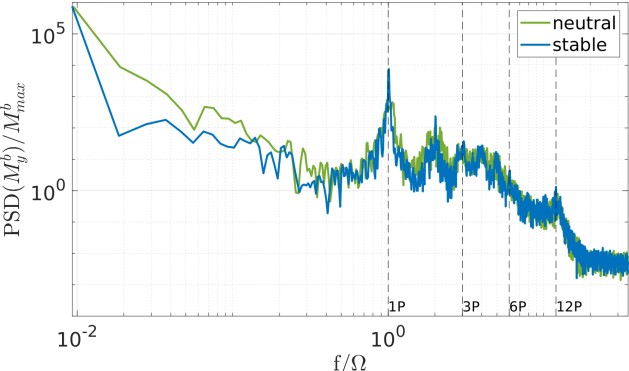

**Figure 24.** Comparison of the blade root bending moments out-of-plane $M_y^b$ for the stable and neutral simulation. The data is normalised by the maximum value of the blade root bending moments and the frequency is normalised by the rotor speed.

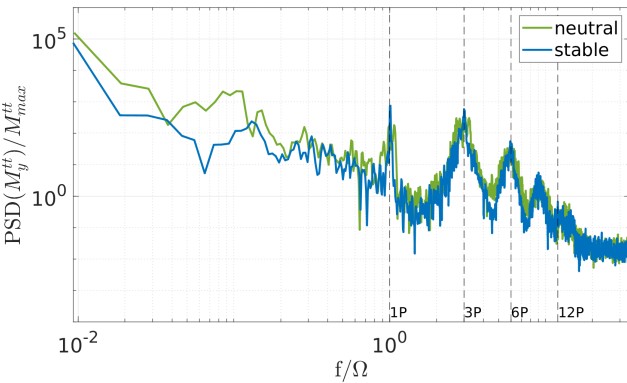

**Figure 25.** Comparison of the tower top fore-aft bending moment $M_y^{tt}$ for the stable and neutral simulation. The data is normalised by the maximum value of the tower top bending moments and the frequency is normalised by the rotor speed.





As can be found in figure 26 the measurement data shows a dependency on the atmospheric stability. Neutral conditions lead to higher power output for the same rotor speed than stable conditions. This behaviour might be explained due to the higher fluctuations caused by higher TI and the therefore higher energy content in the wind. However, the simulations did not reproduce the same dependency, which might be explained by the difference in the turbine control.

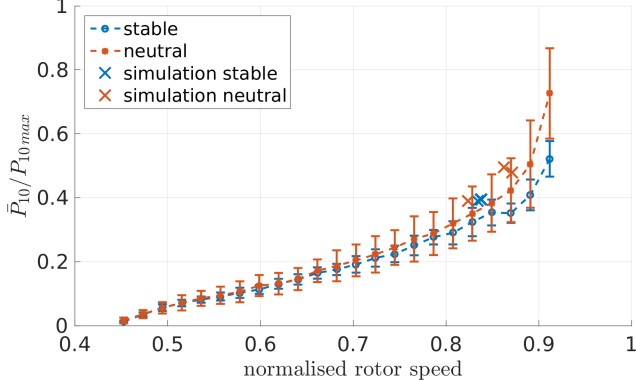

**Figure 26.** Power output, normalised by the maximum measured power, plotted with respect to the rotor speed, normalised with the maximum measured rotor speed with an added offset, for the measurement data in comparison to the simulation data.

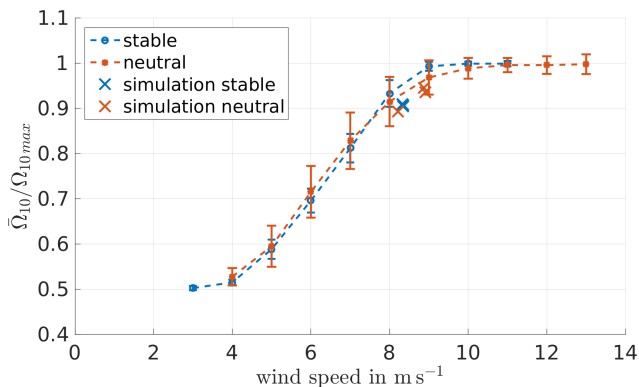

**Figure 27.** Relation between the rotor speed $\Omega$, normalised with the maximum measured rotor speed with an added offset, with respect to the wind speed determined using the measurement data in comparison to the simulation data ($\times$).

## 4  Conclusions

In this paper we presented a new computing routine which combines the advantages of an atmospheric flow simulation using the LES tool PALM and the detailed calculation of the turbine answer by FAST. To quantify the output of the results a comparison to the generic NREL 5 MW turbine and a more extensive comparison to measurement data of a real turbine is shown.





The comparison of the turbine models for the NREL 5 MW turbine showed promising results concerning the quality of the turbine output. Most importantly, it also enabled to make a statement about the computational time of the enhanced coupling.

In the considered cases a saving of computational time of up to 89% could be observed in relation to the equally detailed ALM coupling.

Furthermore, the results for the turbine parameters that are calculated by the new coupling resemble the measured data of the eno114 3.5 MW turbine well. For example the power output is reproduced very well, which is mostly due to the method of taking the wind speed in front of the turbine instead of directly at the rotor area to avoid an overestimation of the power. Also,

the standard variation of the power shows a good resemblance to the measurement data. The parameter reflects the influence of the turbulence in the flow and therefore the stability, which is also present in the simulated results. Keeping in mind, that the simulations were still idealised, i.e. only one homogeneous roughness length and no topography, there is good agreement between the simulated and the measured data.

The blade and tower loads are representative of the measurements in general. Deviations in the aeroelastic simulation model,

especially the tower eigenfrequency, the selected rotor imbalance, the used controller and windspeeds led to slightly different resulting loads compared to the measurements. However, the load spectra still show a very good agreement. Variations due to the atmospheric stability are clearly found. This indicates that the PALM-FAST coupling is suitable to investigate the effects of different atmospheric flows on turbine behaviour.

For future work, a further comparison to measurement data of different situations, such as unstable stratification or in a turbine

wake, is worth considering to further substantiate the results. Examining the performance of the simulations for a turbine in a wake would be a valuable continuation of the current results, as it is not clear how the induction model in particular behaves in a wake. However, due to the reduced computing time, the coupling is basically well suited for carrying out load analyses of a single turbine in a wind farm. As up to now ADM or ADMR has mostly been used in wind farms, since the use of ALM is too computationally intensive due to the required large model domains.

In addition, thanks to the time-saving detailed simulations, there is a multitude of possible applications. Apart from calculating load analyses for wind farms, another possible application is to investigate the relationship between environment and turbine performance in footprint analyses. Furthermore, phenomena in atmospheric flows and their impact on turbine loads can be investigated, such as low level jets.





## Appendix A: Spectra of the loads for the neutral case

The following plots show the blade and tower load spectra for the neutral case.

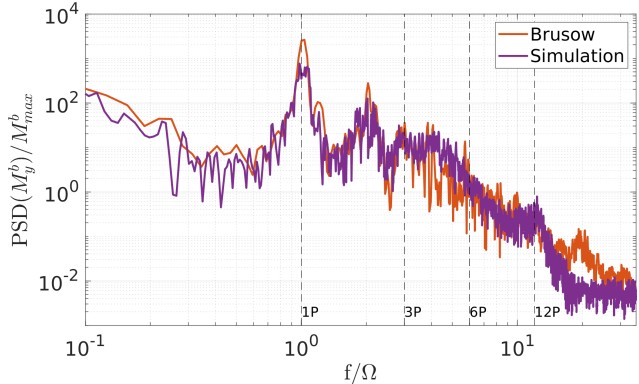

**Figure A1.** Spectrum of the blade root bending out of plane moment $M_y^b$ in comparison to the simulation results (neutral). The data is normalised by the maximum value of the blade root bending moments and the frequency is normalised by the rotor speed.

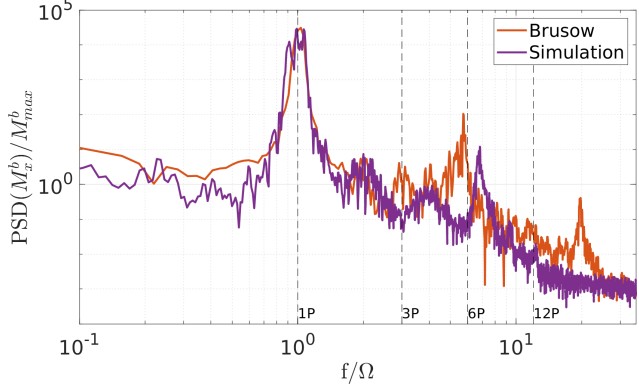

**Figure A2.** Spectrum of the blade root bending in plane moment $M_x^b$ in comparison to the simulation results (neutral). The data is normalised by the maximum value of the blade root bending moments and the frequency is normalised by the rotor speed.

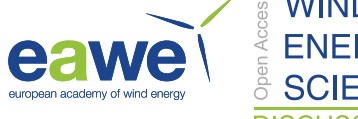



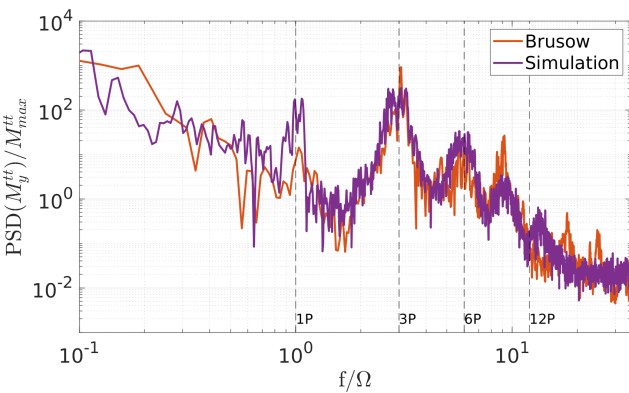

**Figure A3.** Spectrum of the tower top bending moment in fore-to-aft direction $M_y^{tt}$: Comparison of the measurement data to the simulation results (neutral). The data is normalised by the maximum value of the tower top bending moments and the frequency is normalised by the rotor speed.

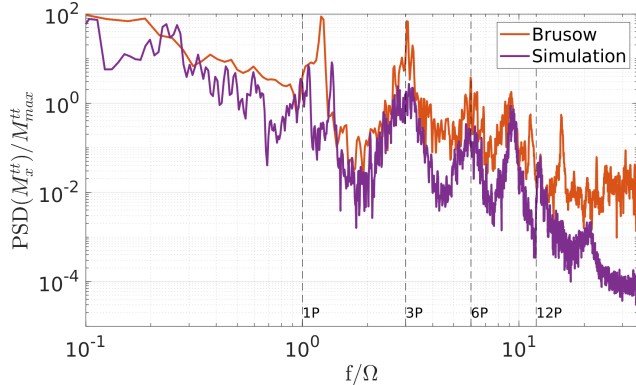

**Figure A4.** Spectrum of the tower top bending moment in side-to-side direction $M_x^{tt}$: Comparison of the measurement data to the simulation results (neutral). The data is normalised by the maximum value of the tower base bending moments and the frequency is normalised by the rotor speed.

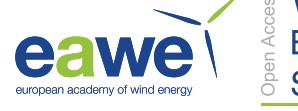
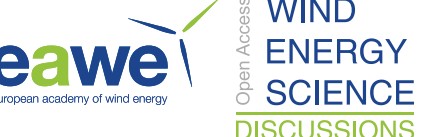

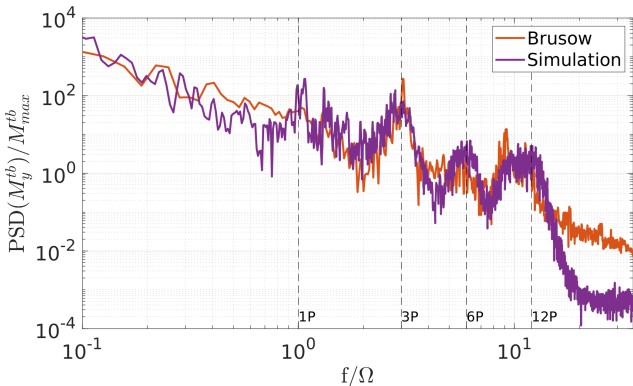

**Figure A5.** Spectrum of the tower base bending moment in fore-to-aft direction $M_y^{tb}$: Comparison of the measurement data to the simulation results (neutral). The data is normalised by the maximum value of the tower top bending moments and the frequency is normalised by the rotor speed.

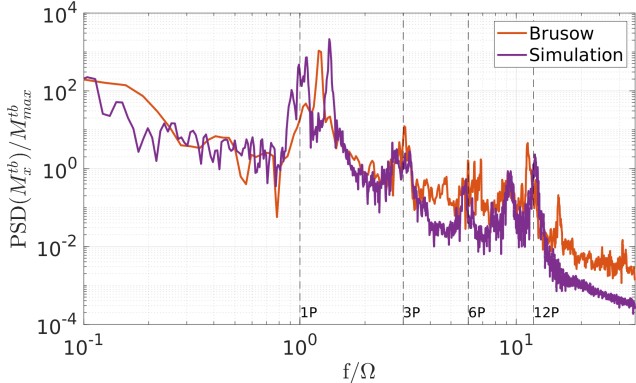

**Figure A6.** Spectrum of the tower base bending moment in side-to-side direction $M_x^{tb}$: Comparison of the measurement data to the simulation results (neutral). The data is normalised by the maximum value of the tower base bending moments and the frequency is normalised by the rotor speed.

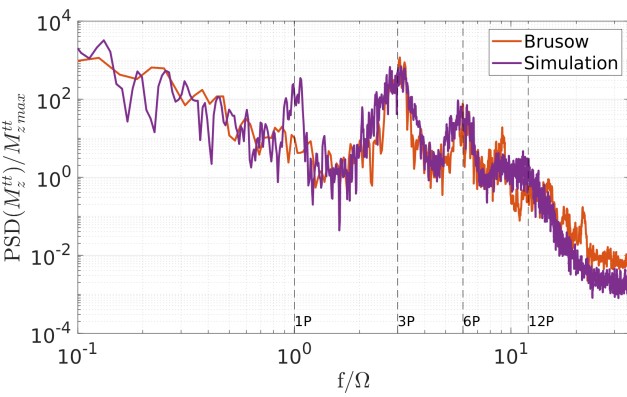

**Figure A7.** Spectrum of the tower top torsion moment $M_z^{tt}$: Comparison of the measurement data to the simulation results (neutral). The data is normalised by the maximum value of the tower top torsion moment and the frequency is normalised by the rotor speed.



## Appendix B: Wind profile comparison for the stable case

Here, the comparison of the wind profiles for the stable case is shown.

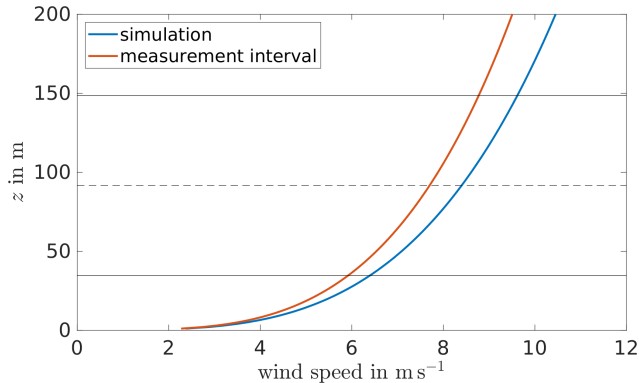

**Figure B1.** Wind profiles, calculated by the shear and wind speed, of the measurement interval and the simulation data used in the comparison of the loads for the stable case. The black lines indicate the rotor area, the dashed line the hub height.

*Data availability.* Simulation and SCADA data of the eno114 3.5 MW turbine are confidential and therefore not available to the public.

*Author contributions.* SK developed the actuator sector method for the PALM-FAST-coupling, performed the simulations and data analy-
445 ses and wrote the paper. GS contributed to acquiring the funding for the work presented in the paper, provided intensive consultation on the development of the method and the scientific analyses. MK provided intensive reviews on the load analyses. LJL provided intensive consultation on the scientific analyses and had a supervising function.

*Competing interests.* The authors declare that they have no conflict of interest.

*Acknowledgements.* The presented work are results of the research projects "WIMS-Cluster" and "ventus efficiens", respectively. "WIMS-
450 Cluster" (FKZ 0324005) was funded by the German Federal Ministry for Economic Affairs and Energy on the basis of a decision by the German Bundestag. "Ventus efficiens" (ZN3024) was funded by the Lower Saxony Ministry for Science and Culture. The computations were performed on the high performance computing system EDDY of the University of Oldenburg funded by the Federal Ministry of Economic Affairs and Energy. We acknowledge the wind turbine manufacturer eno energy for providing SCADA data, the FAST turbine model and their support of the work.



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
