# Peer review of "Validation of a coupled atmospheric-aeroelastic model system for wind turbine power and load calculations"

_Wind Energy Science, 2020_

## Referee Comment (RC1) · Anonymous Referee #1 · 5 Feb 2021

This paper presents an implementation of the actuator surface model using FAST into the LES code PALM. The code is later used to simulate a series of conditions from an experimental campaign.

The manuscript presents interesting results comparing experimental measurements and LES.

Some general comments needed to improve the manuscript are:

1. Extend the literature review, especially for FAST and LES coupling 2. Improve vague language by providing a quantitative assessment (e.g. "enormous reduction of the computing time") 3. Shorten parts of the manuscript (e.g., combine figures and

possibly remove some) 4. Improve the comparison between the turbine models (e.g. add plots for quantities along the blades)

Here are some specific comments to improve the manuscript:

Abstract:

"The comparisons of the simulations to the NREL literature values show very promising results." It is hard to tell what this statement means. I would recommend removing it or talking about something more specific.

"an enormous reduction of the computing time": This statement should be backed up by a number. For example, "our coupling offers a 10X speedup in comparison to similar methods."

Section 2:

Section 2 should be shortened, and the authors should look into the literature. Coupling between FAST and LES has been done for a long time in other codes, see for example: https://www.nrel.gov/docs/fy12osti/53567.pdf https://iopscience.iop.org/article/10.1088/1742-6596/1452/1/012071/meta

Is there a subgrid-scale turbulence model? What are the numerics of the LES code?

Section 3.1:

It is not clear what the difference between the 4 modeling approaches is. A table would be a good way of describing them. The authors only compare power output. To improve the comparison, the authors should look at some quantities along the blade such as velocity, lift, force, etc. Also, how many points along the blade are used?

"The wind speeds at the rotor area are then estimated using the induction model SWIRL of FAST. SWIRL uses the so-called Taylor's frozen turbulence hypothesis (Taylor, 1938) and calculates the induced velocity in axial and tangential direction" It is not clear how this is done. I could not find the "SWIRL" model in the literature. FAST uses

blade element momentum theory to compute induction.

Section 3.2:

What is the power of the reference turbine? The y-label of figure 4 does not seem to be right.

Section 3.2.1:

Figure 5 – This figure needs to be improved. Please add some distances and arrows indicating the wind direction

Section 3.2.3:

There are too many plots in this section. Some plots can be combined into the same figures. For example, the different root bending moments Mx and My can be in the same figure next to each other.

---

## Referee Comment (RC2) · Anonymous Referee #2 · 16 Mar 2021

**1   Overview**

The authors present an innovative for the numerical analysis of wind turbines, combining the modeling of the flow and the assessment of the performance of the machine. Based on its versatility, the authors suggest its use during the design stage, and also during the definition of a wind farm layout. The focus is put on the high fidelity modeling of the flow, based on a Large Eddy Simulation (LES) approach. The wind turbine model relies on an aeroelastic model, referred to as FAST. The coupling between the flow model and the wind turbine model relies on the efficient Actuator Sector Method

(ASM). The authors first show the performance and capabilities of the aforementioned coupling, by means of a relative comparison with other approaches. This is done based on a reference wind turbine (i.e. the NREL 5MW). Finally, the authors include a validation of the approach, where the results of the developed methodology are compared with data acquired in an experimental campaign.

**2  General comments**

The major concerns of the reviewer have to do with the tone and the clarity of the paper. In particular:

- The methodology is not properly described, limiting the impact and reproducibility of the present research. Examples of these are:
  - PALM is only introduced as a "LES-tool". No details about the solver/model are included, apart from the description of the mesh and the boundary conditions. Indeed, no figure of the flow solution is included in the manuscript.
  - FAST is introduced as an aeroelastic tool, but no details about the solver are given. The same comment applies for the FAST model used for the present work.
  - The wind turbine used in the experimental campaign is not properly introduced. For instance "Eno" is not introduced. Is it a wind turbine manufacturer?
  - The developed methodology is presented as a higher fidelity approach, with respect to other techniques (such as RANS for the flow modeling, or the actuator line for the coupling). However, the assumptions and limitations of the proposed methodology are not put into context. For instance, no mention

of rotor-resolved simulations is made in the introduction. In that context, the reviewer is interested in the limitations of the "frozen" windfield (could that compromise the accuracy of the method for particular situations?). However, this issue does not seem to be discussed in the paper.

– Is the ASM approach developed within the framework of the present work?

– Line 164: The authors could give more details around the simulated case (such as omega and pitch).

– Figure 5 could contain more info (such as the relative distance between the wind turbines).

– No details about the controller are given.

• While the manuscript is significantly long, the discussion on the results if often very vague. Examples of these are:

– Figure 3: A discussion of a longer time series, based on statistics/frequency domain, would be necessary before concluding on the capabilities of the method.

– Figure 7: One could also quantify these differences, rather than only providing a qualitative statement.

– Line 407: Discussions around the effect of the wind turbine controller seem to be vague. Could this issue be overcome by a re-tuning of the employed FAST controller?

From the reviewer perspective, the methodology followed to illustrate the performance of the method is insufficient. The comparison exercise around the NREL 5MW is limited to the integrated loads of a very particular case, and a rather poor analysis of a very short time series. It should be emphasized that, contrary to the title of the section, that comparison cannot be classified as a validation (since no experimental results were

considered). The reviewer also misses the presence of the other studied methods in Section 3.2. That could be particularly relevant, given the amount of uncertainty of the comparison itself.

**3  Technical corrections**

- Line 37: "There are different ways to model a turbine, as can be seen in ..." -> Maybe this sentence could be more clear, by replacing by something along the lines "to model the presence of the wind turbine in the flow".

- Line 55: "Computing routine" could be replaced by something more precise, for instance "computing framework".

- Line 153:   would   be   more   readable   if   references   were   grouped (cite{ref1,ref2,ref3}).

- Line 172: "coincides with the expected value" -> "with the value computed by NREL, based on the same FAST model?"

- Line 411: "Turbine answer" -> "turbine response"

---

## Author Comment (AC1) · 7 Sep 2021

**Validation of a coupled atmospheric-aeroelastic model system for wind turbine power and load calculations**

Authors: Sonja Krüger, Gerald Steinfeld, Martin Kraft, and Laura J. Lukassen
DOI: 10.5194/wes-2020-114
* * *
**Authors response to referee comments**

Dear referees, we appreciate the time and effort you spend to give the constructive comments which help to improve our paper. Below we will discuss all comments in detail.

Stating the referees comment (RC) then answering it in the authors comment (AC) and if changes were made in the manuscript, these can be found in the boxes below the authors comment.
* * *
**Authors response to comments from referee #1**

General comments

**RC1** *Extend the literature review, especially for FAST and LES coupling.*

*Improve vague language by providing a quantitative assessment (e.g. "enormous reduction of the computing time").*

*Shorten parts of the manuscript (e.g., combine figures and possibly remove some).*

*Improve the comparison between the turbine models (e.g. add plots for quantities along the blades)*

**AC** Thank you for these general remarks, we will address them individually below in the specific comments.

Specific comments

**RC2** *Abstract: "The comparisons of the simulations to the NREL literature values show very promising results." It is hard to tell what this statement means. I would recommend removing it or talking about something more specific.*

**AC** With this statement we wanted to indicate that in our first approach in a comparison with the NREL turbine promising results were made. However, no details were given here in order to keep the abstract simple and short. We agree that this way there may not be enough information given in the abstract, so the sentence was adjusted to clarify the statement.

> p. 1, ll. 10f: The basic test of the coupling with the NREL 5 MW turbine shows that the power curve obtained is very close to the one when using FAST alone.

**RC3** *Abstract: "an enormous reduction of the computing time": This statement should be backed up by a number. For example, "our coupling offers a 10X speedup in comparison to similar methods."*

**AC** We only have numbers for certain cases, but we agree that a specification improves the abstract here. Therefore, to specify the statement we added a number without going into too much detail.

> p. 1, ll. 12f: Additionally, our coupling offers an enormous reduction of the computing time, in one of our cases by 89%, and at the same time an extensive output of turbine data.

**RC4** *Section 2(?): Section 2(?) should be shortened, and the authors should look into the literature. Coupling between FAST and LES has been done for a long time in other codes, see for example: https://www.nrel.gov/docs/fy12osti/53567.pdf https://iopscience.iop.org/article/10.1088/1742-6596/1452/1/012071/meta.*

 **AC** Since this comment is aimed at the literature review, we assume that it is actually concerned with Section 1. This section aims to give an introduction to the topic to a wide range of readers with different backgrounds. We agree that the overview was too broad in some parts and not deep enough in others. We have addressed this by omitting some overly detailed information and adding more detail on LES/FAST couplings, as suggested.

**RC5** *Section 2: Is there a subgrid-scale turbulence model? What are the numerics of the LES code?*

 **AC** Yes, the LES code PALM offers different subgrid-scale (SGS) turbulence models. The default model was used, which in the revised manuscript was included in the description of the set-up in section 3. In addition, the description has been expanded to include information about the time stepping and advection scheme and the pressure solver.

> p. 7, ll. 179f: Additionally, the surface condition is set to a free slip condition. PALM offers different possibilities for the subgrid-scale turbulence closure. For the simulations mentioned in this work the default model was used which is a modified version of Deardorff's subgrid-scale model (Deardorff, 1980), as mentioned in (Moeng and Wyngaard, 1988) and (Saiki et al., 2000). The time stepping and advection schemes were used in the default settings as well, which is a third order Runge-Kutta scheme (Williamson, 1980; Baldauf, 2008) for time stepping and a fifth order upwind scheme, based on (Wicker and Skamarock, 2002), for the advection. The pressure solver was set to the multigrid option (Uhlenbrock, 2001).

**RC6** *Section 3.1: It is not clear what the difference between the 4 modeling approaches is. A table would be a good way of describing them.*

 **AC** A table giving an overview of the models is helpful to understand the differences more quickly. Therefore we have added table 1.

**Table 1.** Overview of the turbine models that were used in the comparisons. The new enhanced coupling method is ASM, the respective time steps in PALM and FAST are denoted as $\Delta t_P$ and $\Delta t_F$ respectively and the inflow wind speed as $U$.

| Name | Time step | Wind speed information | Rotor model in PALM |
|---|---|---|---|
| ALM | coupled time step: $\Delta t_P = \Delta t_F$ | $U$ taken at positions of rotor blade elements in PALM | line |
| ASM w/o SWIRL | decoupled time step: $\Delta t_P = n \cdot \Delta t_F$ | $U$ taken at positions of rotor blade elements in PALM, from a frozen wind field | sector |
| ASM | decoupled time step: $\Delta t_P = n \cdot \Delta t_F$ | $U$ taken upstream of the rotor blade positions in PALM and use of the induction model SWIRL of FAST | sector |
| FAST | only FAST $\Delta t_F$ | steady wind case in FAST, no LES | - |

**RC7** *Section 3.1: The authors only compare power output. To improve the comparison, the authors should look at some quantities along the blade such as velocity, lift, force, etc. Also, how many points along the blade are used?*

   **AC**  Thank you for this suggestion. We have added a comparison of dynamic pressure, lift and drag coefficient as well as the angle of attack along the blade for the laminar simulation of the NREL 5 MW turbine. Additionally, we had to renew the ALM curve in figure 3 as the previous one had a different input in comparison to the other methods, however, this does not affect the results.

> p. 8, ll. 198f: A comparison of quantities along the 62 blade nodes show a difference between the methods using wind speeds at the rotor blade positions (ALM and ASM w/o SWIRL) and the two methods using a different inflow, namely ASM and FAST (figures can be seen in Appendix A). The distribution of the angle of attack shows a smoothed curve for the ALM and ASM w/o SWIRL, which is due to the smearing of the forces around the rotor blades. On the other hand ASM and FAST show a choppy curve due to the different airfoil profiles along the blade, here, it can be seen at which position a change of an airfoil profile and twist angle along the blade is predefined in the NREL model. These differences are transferred to the lift and drag coefficients. Additionally, for dynamic pressure, it can be observed that ALM and ASM w/o SWIRL overestimate the dynamic pressure at the blade tips and slightly underestimate it at the hub compared to FAST and ASM. These observations suggest that the smearing of the forces along the blade has a great influence on the lift and drag properties and thus the turbine response.

**RC8** *Section 3.1: "The wind speeds at the rotor area are then estimated using the induction model SWIRL of FAST. SWIRL uses the so-called Taylor's frozen turbulence hypothesis (Taylor, 1938) and calculates the induced velocity in axial and tangential direction". It is not clear how this is done. I could not find the "SWIRL" model in the literature. FAST uses blade element momentum theory to compute induction.*

   **AC**  The documentation of FAST is incomplete regarding SWIRL. SWIRL is based on the calculation of the induction factors according to (Harman, 1994), which is only documented in the source code of FAST. We have clarified the respective sentence.

> p. 5, ll. 141f: The wind speeds at the rotor area are then estimated using the induction model SWIRL of FAST. SWIRL uses the so-called Taylor's frozen turbulence hypothesis (Taylor, 1938) and calculates the induced velocity in axial and tangential direction. In (Moriarty and Hansen, 2005) the Aerodyn model of FAST is described, including the blade-element momentum (BEM) theory to compute the induction. The calculation of the induction factors when using SWIRL is based on (Harman, 1994). With enabling SWIRL we assume, that the turbulent structures in the wind field do not change while moving to the turbine.

**RC9** *Section 3.2: What is the power of the reference turbine? The y-label of figure 4 does not seem to be right.*

   **AC**  Yes, you are correct, there was a mistake in the y-scale of figure 4. Figure 4 shows the correct scale.

[Figure]

**Figure 4.** Comparison of different simulation methods for the generator power of the eno114 3.5 MW turbine in a laminar case with a wind speed of $8\,\mathrm{ms^{-1}}$, normalised by the respective value of the eno power curve at $8\,\mathrm{ms^{-1}}$.

**RC10** *Section 3.2.1: Figure 5 – This figure needs to be improved. Please add some distances and arrows indicating the wind direction.*

    **AC** Thank you for the hint about the missing distances, we have adjusted the drawing accordingly (see figure 5). However, an indication of a specific wind direction may not be useful, since measurement data from all wind directions was available and filtered in such a way that neither the forest nor the measurement mast influenced the used data. The wind directions remaining after the filtering are indicated in red. We have modified the figure caption accordingly.

[Figure]

**Figure 5.** Schematic of the measurement site in Brusow. The remaining wind directions in the measurement data, after filtering, are indicated in red; $D$ is the turbine diameter, here $D$=114.9 m.

**RC11** *Section 3.2.3: There are too many plots in this section. Some plots can be combined into the same figures. For example, the different root bending moments Mx and My can be in the same figure next to each other.*

    **AC** There is a large number of plots in this section. However, they all show different information. The suggestion to combine some plots into one figure was very helpful and was implemented where possible.

**Authors response to comments from referee #2**

General comments

**RC1** *The methodology is not properly described, limiting the impact and reproducibility of the present research. Examples of these are:*

*a) PALM is only introduced as a "LES-tool". No details about the solver/model are included, apart from the description of the mesh and the boundary conditions. Indeed, no figure of the flow solution is included in the manuscript.*

*b) FAST is introduced as an aeroelastic tool, but no details about the solver are given. The same comment applies for the FAST model used for the present work.*

*c) The wind turbine used in the experimental campaign is not properly introduced. For instance "Eno" is not introduced. Is it a wind turbine manufacturer?*

*d) The developed methodology is presented as a higher fidelity approach, with respect to other techniques (such as RANS for the flow modeling, or the actuator line for the coupling). However, the assumptions and limitations of the proposed methodology are not put into context. For instance, no mention of rotor-resolved simulations is made in the introduction. In that context, the reviewer is interested in the limitations of the "frozen" windfield (could that compromise the accuracy of the method for particular situations?). However, this issue does not seem to be discussed in the paper.*

*e) Is the ASM approach developed within the framework of the present work?*

*f) Line 164: The authors could give more details around the simulated case (such as omega and pitch).*

*g) Figure 5 could contain more info (such as the relative distance between the wind turbines).*

*h) No details about the controller are given.*

**AC**

a) Thank you for that hint. As PALM is a complex model we referenced two PALM papers with further, detailed information. However, we agree that more detailed information can improve the quality of the paper, so we added some information on PALM in general and on the settings used by us in the description of the simulation set-up, i.e. the subgrid-scale model, the time stepping scheme, the advection scheme and the pressure solver.

> p. 3, ll. 87f: PALM enables the simulation of an atmospheric flow for a wide range of different situations, like e.g. different stabilities using heating or cooling of the surface. It is based on the non-hydrostatic, filtered, incompressible Navier-Stokes equations in Boussinesq-approximated form and has seven prognostic quantities: the wind speed on a cartesian grid $u$, $v$, $w$, the potential temperature $\Theta$, the water vapor mixing ratio $q_v$, a passive scalar $s$ and the subgrid-scale turbulent kinetic energy $e$. The domain is divided into equidistant cells in horizontal direction, stretching of the cells is possible in vertical direction. To define the position of the quantities the Arakawa staggered C-grid (Harlow and Welch, 1965; Arakawa and Lamb, 1977) is used.
>
> p. 7, ll. 179f: PALM offers different possibilities for the subgrid-scale turbulence closure. For the simulations mentioned in this work the default model was used which is a modified version of Deardorff's subgrid-scale model (Deardorff, 1980), as mentioned in (Moeng and Wyngaard, 1988) and (Saiki et al., 2000). The time stepping and advection schemes were used in the default settings as well, which is a third order Runge-Kutta scheme (Williamson, 1980; Baldauf, 2008) for time stepping and a fifth order upwind scheme, based on (Wicker and Skamarock, 2002), for the advection. The pressure solver was set to the multigrid option (Uhlenbrock, 2001).

b) As far as FAST is concerned, most of the used set-up is described in section 3.2.2. However, more information about the AeroDyn module has been added.

> section 3.2.2, Ende: In AeroDyn the Beddoes-Leishman dynamic stall model, based on (Leishman and Beddoes, 1989) and the "Equil" option, a BEM model, for the inflow was selected. Additionally, the tip-loss and hub-loss models were enabled and set to the Prandtl tip loss model (Prandtl and Betz, 1927).

c) Yes, eno is a wind turbine manufacturer. We added the information when first mentioning details about the turbine, further information can be found in the referenced data sheet.

> p. 8, ll. 202f: Measurement data of an eno114 3.5 MW turbine, manufactured by eno (eno energy, 2019), with a hub height of 92 m and the corresponding FAST turbine model are used for further investigations.

d) We have made additions concerning this in the introduction and in the conclusion.

> p. 2 ll. 59f: A fully resolved wind turbine simulation can lead to the same or even more detailed output, but is far more computationally intensive than the presented framework.
> p. 26 ll. 463f: In the current work, the constraints of the frozen wind field, e.g. the assumption of Taylor's frozen turbulence hypothesis, does not limit the outcome, as in the current simulations, the statistics of the flow are not subject to varying wind conditions. However, there are also situations where the hypothesis will reach its limits, e.g. with temporally variable wind fields or changing wind direction. The case of a turbine in a wake also needs further investigation, as the recovery of the wake in the frozen wind field has not been considered so far.

e) This particular sector method was developed within the framework of the present work. However, different sector methods were already existent, like e.g. the version published in (Storey et al., 2015), as mentioned in the introduction.

> p. 3 ll. 65f: We developed one variation of an Actuator Sector Method (ASM), where the blade movement is described as a segment of a circle. This allows for a larger time step in PALM than in FAST as the movement of the blade during that time step is captured in the area of the sector.

f) As the controller of the NREL 5 MW turbine was used as it is available from NREL and described in (Jonkman et al., 2009b), the information on the behaviour of the NREL 5 MW turbine is already described in more detail in other publications. However, the information, that the used controller does not deviate from the known turbine response was added to the paper.

> p. 7, ll. 185f: The standard controller of the NREL turbine is employed as described in (Jonkman et al., 2009b), which means that at the prevailing wind speeds no pitching of the blades is enabled.

g) Yes, figure 5 needed to be improved, the modified version can be seen in figure 5.

h) The NREL turbine uses its standard controller, which is included when downloading the turbine model and is described in (Jonkman et al., 2009b). The controller of the eno turbine is confidential and was not made public to us, only a .dll file was used.

p. 16, ll. 315f: The turbine model of the eno114 3.5 MW turbine for FAST was provided by eno, including structural information and a pitch, a speed and a yaw control in the format of a Bladed .dll file, which was not accessible to us.

**RC2** *While the manuscript is significantly long, the discussion on the results if often very vague. Examples of these are:*

*a) Figure 3: A discussion of a longer time series, based on statistics/frequency domain, would be necessary before concluding on the capabilities of the method.*

*b) Figure 7: One could also quantify these differences, rather than only providing a qualitative statement.*

*c) Line 407: Discussions around the effect of the wind turbine controller seem to be vague. Could this issue be overcome by a re-tuning of the employed FAST controller?*

**AC**

a) The above mentioned time series was not intended for a deeper analysis of frequencies or statistics. The intention for using the NREL turbine in a turbulent simulation was to examine whether the method of using wind speed in front of the turbine and assuming Taylor's frozen turbulence hypothesis would lead to a useful result. In particular, it was intended to investigate whether the turbulence can be represented in a comparable way to other models in the power output. The used simulations were stationary due to their preruns and therefore a good starting point for a comparison. When running the different models for a longer time, the results will drift apart at some point due to the different turbine responses, as interferences in the wind field are transported upstream as well due to the pressure solver. However, this was not the objective of this particular simulation.

b) We have replaced the plot of the power curve with figure 7, here the differences can be seen more clearly and the deviation between the stabilities can be read directly from the plot. The new plot shows the power curves normalised by the respective eno power value, which is determined in simulations using FAST and a laminar situation.

[Figure]

**Figure 7.** Power data determined from the measurement data for May/June 2017, normalised by the corresponding power of the eno power curve determined by FAST in laminar conditions, for different stabilities (determined from eddy-covariance data).

c) As mentioned above we did not have access to the eno controller and therefore no information on how the controller influences the results. However, on further dealing with the above mentioned paragraph we now think that the differences can be explained by the limited turbulence intensity (TI) variation. The simulations only cover a small TI range, the differences between the runs are not very large. In contrast, the measured data show a greater variability of the TI, which can lead to a higher power output in the neutral case than in the stable case at the same rotor speed, whereas this is not visible in the simulation.

> p. 24, ll. 440f: However, the simulations did not reproduce the same dependency, which might be explained by the limited variability of the TI in comparison to the measurement data. As can be seen in figure 14 the simulations cover the lower limit of the TI in the respective wind speed.

**RC3** *From the reviewer perspective, the methodology followed to illustrate the performance of the method is insufficient. The comparison exercise around the NREL 5MW is limited to the integrated loads of a very particular case, and a rather poor analysis of a very short time series. It should be emphasized that, contrary to the title of the section, that comparison cannot be classified as a validation (since no experimental results were considered). The reviewer also misses the presence of the other studied methods in Section 3.2. That could be particularly relevant, given the amount of uncertainty of the comparison itself.*

    **AC** Yes, thank you for this suggestion, section 3.1 discussing the NREL turbine is misleading when using the title "Validation [...]". The section was not intended for a sole validation, but was seen as a trial of the general performance of the enhanced coupling. As we were using wind speeds in front of the turbine, assumed Taylor's hypothesis and used a force projection of a sector, in contrast to the previously employed methods, an evaluation of the general performance was our first intention. The purpose was to answer whether in general the output seemed reasonable, whether the coupling would work in a turbulent flow and whether the computational time is reduced. Once, the basis performance of the enhanced coupling was satisfying an extensive validation with measurement data was done. A further analysis, including more thorough simulations, with the NREL turbine did not appear to be very effective, due to the lack of measurement data of the NREL turbine.

Nevertheless, this was not made clear in this section and needed to be clarified in the text.

In section 3.2 the focus was set on only the ASM method as the intention was to validate the simulation results and not compare the behaviour of different methods. Also, the other methods, that were employed in section 3.1 each have a drawback, i.e. the computational resources for the ALM are very high and due to limited memory and computational time further simulations were not carried out. The ASM without the induction model SWIRL showed already in section 3.1 that the results are not as good as with the induction model, therefore it did not seem feasible to follow up this method. Lastly, using FAST without a coupling to PALM needs a different source for a turbulent windfield. Generating one with a different tool than PALM would complicate a possible comparison, as this would add a new variable. Furthermore, the result of a FAST simulation with a different inflow would not be helpful in validating the enhanced coupling method.

> p. 6, ll. 164f: As this is a generic turbine, no comparison with measured data is possible. But the availability of the turbine data allows an evaluation of our enhanced coupling method, also in terms of turbulent flows. Additionally, the availability of the turbine data offers the opportunity to compare different methods and their computational resources.
>
> p. 8, ll. 226f: Through these simple simulations it can be seen that the sector methods offer savings in the computing time in comparison to the ALM. However, the ASM w/o SWIRL does not provide the expected results. Therefore, it is considered useful to compare the ASM with measurement data in the following.

Specific comments

**RC4** *Line 37: "There are different ways to model a turbine, as can be seen in ..." -> Maybe this sentence could be more clear, by replacing by something along the lines "to model the presence of the wind turbine in the flow".*

    **AC** Yes, the phrasing was a bit too vague. Thank you for the suggestion.

p. 2, ll. 29f: There are different ways to model the presence of a wind turbine in the flow, as can be seen in e.g. (Witha et al., 2014) and (Wu and Porté-Agel, 2013).

**RC5**  *Line 55: "Computing routine" could be replaced by something more precise, for instance "computing framework".*

   **AC**  Computing routing might be too general for a simulation method, the suggested alternative seems to describe the program better and was changed.

p. 2, ll. 56f: To address the problem of losing information of either the turbine or the flow and provide a reliable tool, we present a newly developed computing framework here, with which it is possible to calculate LES in combination with a well resolved turbine model, i.e. apart from the power output also quantities for the blades and along the blades are available.

**RC6**  *Line 153: would be more readable if references were grouped (citeref1,ref2,ref3).*

   **AC**  Thank you for that suggestion.

p. 6, ll. 160f: The NREL 5 MW turbine (Jonkman et al., 2009b) is a generic turbine which has been used extensively in simulations (Churchfield et al., 2012; Storey et al., 2013, 2015; Vollmer et al., 2016; Sathe et al., 2013; Lee et al., 2012).

**RC7**  *Line 172: "coincides with the expected value" -> "with the value computed by NREL, based on the same FAST model?"*

   **AC**  Yes, all four models use the same FAST model of the NREL turbine, but different ways of retrieving the wind speed information or distributing the forces back in the flow field. This is now summarised in table 1. Related to specifically the method of using solely FAST it is correct, that it is the same calculation with the same tool and model as presented in (Jonkman et al., 2009b). As there are quite a number of possible input variations in FAST it seemed sensible to check whether our simulation set-up coincides with the one used in (Jonkman et al., 2009b).

p. 7, ll. 188f: The result calculated by FAST coincides with the value, as published by NREL (Jonkman et al., 2009b), based on the same FAST model.

**RC8**  *Line 411: "Turbine answer" -> "turbine response"*

   **AC**  Thank you for pointing that out.

p. 25, ll. 444f: In this paper we presented a new computing framework which combines the advantages of an atmospheric flow simulation using the LES tool PALM and the detailed calculation of the turbine response by FAST.

**References**

Arakawa, U. and Lamb, V.: Computational design of the basic dynamical processes of the UCLA general circulation model. in: General Circulation Models of the Atmosphere, Methods in Computational Physics, 17, 173–265, 1977.

Baldauf, M.: Stability analysis for linear discretisations of the advection equation with Runge-Kutta time integration, Journal of Computational Physics, 227, 6638–6659, 2008.

Churchfield, M., Lee, S., Michalakes, J., and Moriarty, P. J.: A numerical study of the effects of atmospheric and wake turbulence on wind turbine dynamics, Journal of Turbulence, 13, 1–32, 2012.

Deardorff, J. W.: Stratocumulus-capped mixed layers derived from a three-dimensional model, Boundary-Layer Meteorology, 18, 495–527, 1980.

eno energy: eno114 3.5, https://www.eno-energy.com/fileadmin/downloads/datenblatt/ENO_114_ENG_Datenblatt_AS.pdf, 2019.

Harlow, F. and Welch, J.: Numerical calculation of time-dependent viscous incompressible flow of fluid with free surface, The Physics of Fluids, 8, 2182–2189, 1965.

Harman, C.: PROPX: Definitions, Derivations, and Data Flow, 1994.

Jonkman, J., Butterfield, S., Musial, W., and Scott, G.: Definition of a 5-MW Reference Wind Turbine for Offshore System Development, Tech. Rep. NREL/TP-500-38060, National Renewable Energy Laboratory, 2009b.

Lee, S., Churchfield, M., Moriarty, P., Jonkman, J., and Michalakes, J.: Atmospheric and Wake Turbulence Impacts on Wind Turbine Fatigue Loading - Preprint, 50th AIAA Aerospace Sciences Meeting including the New Horizons Forum and Aerospace Exposition, 2012.

Leishman, J. and Beddoes, T.: A Semi-Empirical Model for Dynamic Stall, Journal of the American Helicopter Society, 34, 3–17, 1989.

Moeng, C. and Wyngaard, J. C.: Spectral analysis of large-eddy simulations of the convective boundary layer, Journal of the Atmospheric Sciences, 45, 3573–3587, 1988.

Moriarty, P. J. and Hansen, A. C.: AeroDyn Theory Manual, Tech. Rep. NREL/EL-500-36881, National Renewable Energy Laboratory, 2005.

Prandtl, L. and Betz, A.: Vier Abhandlungen zur Hydrodynamik und Aerodynamik, Göttinger Nachrichten, pp. 88–92, 1927.

Saiki, E., Moeng, C., and Sullivan, P.: Large-eddy simulation of the stably stratified planetary boundary layer, Boundary-Layer Meteorology, 95, 1–30, 2000.

Sathe, A., Mann, J., Barlas, T., Bierbooms, W. A. A. M., and van Bussel, G. J. W.: Influence of atmospheric stability on wind turbine loads, Wind Energy, 16, 1013–1032, 2013.

Storey, R. C., Norris, S. E., Stol, K. A., and Cater, J. E.: Large eddy simulation of dynamically controlled wind turbines in an offshore environment, Wind Energy, 16, 845–864, 2013.

Storey, R. C., Norris, S. E., and Cater, J. E.: An actuator sector method for efficient transient wind turbine simulation, Wind Energy, 18, 699–711, 2015.

Uhlenbrock, J.: Entwicklung eines Multigrid-Verfahrens zur Lösung elliptischer Differentialgleichungen auf Massivparallelrechnern und sein Einsatz im LES-Modell PALM, Master's thesis, Institute of Meteorology and Climatology, Leibniz University Hanover, 2001.

Vollmer, L., Steinfeld, G., Heinemann, D., and Kühn, M.: Estimating the wake deflection downstream of a wind turbine in different atmospheric stabilities: an LES study, Wind Energy Science, 1, 129–141, 2016.

Wicker, L. and Skamarock, W.: Time-Splitting Methods for Elastic Models Using Forward Time Schemes, Monthly Weather Review, 130, 2088–2097, 2002.

Williamson, J.: Low-storage Runge-Kutta schemes, Journal of Computational Physics, 35, 48–56, 1980.

Witha, B., Steinfeld, G., and Heinemann, D.: High-Resolution Offshore Wake Simulations with the LES Model PALM, Hölling M., Peinke J., Ivanell S. (eds) Wind Energy - Impact of Turbulence, 2, Springer, Berlin, Heidelberg, 2014.

Wu, Y.-T. and Porté-Agel, F.: Simulation of Turbulent Flow Inside and Above Wind Farms: Model Validation and Layout Effects, Boundary-Layer Meteorology, 146, 181–205, 2013.

---

## Author Response (AR2)

**Validation of a coupled atmospheric-aeroelastic model system for wind turbine power and load calculations**

Authors: Sonja Krüger, Gerald Steinfeld, Martin Kraft, and Laura J. Lukassen

DOI: 10.5194/wes-2020-114
* * *
**Authors response to referee comments**

Dear referees, we appreciate the time and effort you spend to give constructive comments. Following your comments and suggestions lead to a considerable improvement of our manuscript. Below we will refer to all of your comments in detail.
The referees comments (RC) are listed with the authors comments (AC). If changes have been made in the manuscript, these can be found in the boxes below the authors comment.
* * *
**Authors response to comments from referee #2**

Major comments

**RC1** *L67: It is mentioned that the implemented methodology can be seen as an enhancement with respect to Storey (2015). However, this is not further detailed in the manuscript. Is there any novelty, rather than the implementation on a different set of solvers, on the computational methods used?. Whether is the case or not, it should be clearly stated in the text.*

**AC**  In the paper (Storey et al., 2015) an Actuator Sector Method is presented. The paper describes the approach of using a sector as a compromise between an Actuator Line and an Actuator Disk method, just as our approach, but the realisation on how the forces are projected as a sector shape into the flow are different to the PALM-FAST coupling and Storey did not present an analysis of loads obtained by applying his coupling approach. Therefore, it is important to mention the paper of Storey, but we do not consider our method an enhancement with respect to Storey. The relevant phrase in line 67 states that: "A similar method is suggested in (Storey et al., 2015), where an ASM is tested in simulations." To clarify the differences and similarities we have made some additions to this phrase.

> p.3, ll. 67f: A similar method is suggested in (Storey et al., 2015), where an ASM is tested in simulations. In order to combine the respective advantages of an ALM and an ADM (Storey et al., 2015) presents a sector method, that uses a different approach of projecting the forces into the flow than is presented in this paper.

**RC2** *L148: "this can be resolved in the postprocessing of the results by shifting the results in time" -> would this imply a certain level of approximation? In other words, is it a "model" that could be applied in the post-processing step? In that case it would more an "estimation" rather than a "solution" itself.*

**AC**  In order to be able to compare different turbine models, as in figure 3, the time shift of the ASM needs to be addressed. To resolve this we shifted the results in time and mentioned this as a possible step in the postprocessing of the data. However, at this position (ll. 148f) in the paper this might be confusing. Therefore, we have moved and modified the information to the section where the shifting of the results in figure 3 is mentioned.

> p.8, ll. 217f: Therefore, when comparing the turbine output the result of the ASM simulation is shifted by 34 s for a better comparison to the other results. This does not affect the statistics but is a simple method to make the

time series obtained from the different models comparable to each other. A model or tool that automatically fixes this time shift is not included in the current version of the coupling.

**RC3** *L172: "For FAST on its own, the inflow wind option "steady wind conditions" is used.": this should be re-phrased in conceptual terms, rather than as the naming used for the FAST software. Does it simply mean laminar inflow? The reviewer believes that this should have been consistent with the setup of the other methods, for the laminar comparison. So that it is something related to the load case itself, rather than to FAST alone.*

**AC** To mention the "steady wind conditions" option seemed relevant, as multiple options are possible for the inflow in FAST. However, it might not have been clear enough, that by choosing this, the simulation is comparable to the others as the inflow is laminar as well.

> p.6, ll. 171f: As fourth method, just in the laminar case, FAST on its own is used (denoted as FAST). For FAST on its own, the inflow wind option is set to match the PALM simulations, i.e. the power law variables are set to a wind speed of 8 m/s constant with time and with height.

**RC4** *L173: "To evaluate the different methods, at first, a laminar case with the same wind speed over height is considered.": could be interesting to give more details about the specific profile that was used, e.g. relating it to a shear factor.*

**AC** The considered laminar case had a constant wind speed with height, i.e. zero vertical gradient of the streamwise velocity. As this might have been not made clear enough, this was added in the text.

> p.6, ll. 174f: To evaluate the different methods, at first, a laminar case with a constant wind speed with height, i.e. zero vertical gradient of the streamwise velocity, is considered.

**RC5** *L177: " However, no differences in the results ": This could be quantified, in order to be more precise. Maybe the authors can show the relative difference of a targeted quantity, just as an example.*

**AC** Different sized model areas were tested and the wind speed and turbine response were compared, no significant differences were seen here and therefore the smaller one was chosen for further simulations.

> p.7, ll. 178f: However, no significant differences in the conditions of the flow in the turbulent case (i.e. a deviation of 2% in the wind speed at 92 m) or the turbine output, were detected and therefore the smaller model domain was used for the simulations.

**RC6** *L188: "The result calculated by FAST coincides with the value, as published by NREL (Jonkman et al., 2009b), based on the same FAST model.": The reviewer wonders why is this comment important. There is no quantification of the relative differences, and from an academic perspective is probably not that relevant. It is assumed that a simulation of the same inflow conditions and with the same code, will lead to the same results. If the comment had to do with software versioning, then it should be clear in the text. If it was more related to potential discrepancies while building the aeroelastic model, the possible origin of those should be mentioned.*

**AC** As FAST has numerous possibilities to be set-up, this information was considered as important. However, as the comparability to NREL literature values has no impact on the comparison to the PALM-FAST coupling we have now decided to omit this statement as it seems to be more confusing than helpful.

**RC7** *L207: "A turbulent case is calculated as well.": Could be interesting to lift this comment up in the text, while introducing the load cases. Note that several comments on the FAST inflow have been already made at this point, and the reader could lack of context.*

    **AC** Thank you for pointing this out. In order to make the workflow more comprehensible, we have added the information earlier in the text and adapted it where necessary.

> p.6, ll. 163f: As this is a generic turbine, no comparison with measured data is possible. But the availability of the turbine data allows an evaluation of our enhanced coupling method, also in terms of turbulent flows. Additionally, the availability of the turbine data offers the opportunity to compare different methods and their computational resources. Therefore, two cases were considered, firstly a laminar and secondly a turbulent flow. [...]
> p.6, ll. 171f: As fourth method, just in the laminar case, FAST on its own is used (denoted as FAST). [...]
> p.8, ll. 208f: As a second case a turbulent flow is calculated.

**RC8** *L219: "Also, roughly the same peaks and therefore structures of the flow are present in the ASM results. This implies, that the coupling works in a turbulent environment as well.": The reviewer thinks that there is not enough evidence for such a statement. The phrase "the coupling works" is very vague, as it relies on a qualitative observation. There is room, for instance, for an implementation bug that could eventually shift the loading spectra during the coupling, or that accounts for a bad projection. To comment on the flow structures a qualitative measure should be used.*

    **AC** Thank you for pointing this out. As mentioned in our previous response this was a first test of the coupling in order to see whether this might lead to a successful model. For the NREL turbine no deeper analysis was intended due to the lack of measurement data. However, you are correct, that the conclusion we drew from this short test case went too far. Therefore, we adjusted the statement to only relate to the power output.

> p.8, ll. 222f: Also, roughly the same peaks and therefore structures of the flow are present in the ASM results. This indicates that the coupling also works in a turbulent environment insofar as the turbulent structures are reflected in the power output.

**RC9** *L234: " The reference power curve is obtained from stand-alone FAST runs, with a laminar inflow. The FAST turbine model is provided by eno. The calculated reference power curve coincides well with the published power curve of eno (eno energy, 2019).": the reviewer wonders what is the added value of this statement. There is no plot shown, and no discussion on how the power curves were computed from the manufacturer side.*

    **AC** Reproducing the power curve of the manufacturer shows, that the coupling and in particular the turbine model, match the manufacturer's data. This might seem of little importance, but the manufacturer's controller was not readable (only available in binary form) to us and a wide range of input options are possible in FAST, hence it was a valuable check. A plot of the matching power curve was not seen as more informative than the statement itself.
Neither information on how the manufacturer calculated the power curve nor the data, apart from the published curve, was available to us.

> p.9, ll. 238f: The reference power curve is obtained from stand-alone FAST runs, with a laminar inflow. The FAST turbine model is provided by eno, but the source code of the turbine controller was not available to us, only an executable file was provided. The calculated reference power curve coincides well with the published power curve of eno (eno energy, 2019), without figure. Of the published power curve no further information on the computation or data is available and therefore no comparative plot is possible.

**RC10**  *L360: "The results of the simulations correspond well with the measurement data": In the opinion of the reviewer, there is not enough evidence to support this statement.*

    **AC**  In order to compare the atmospheric conditions between simulations and measurement data, the wind speed, the shear and the turbulence intensity were used. The simulations provide situations that could have been measured at that site, see figure 13. For the comparison of the corresponding loads the measurement data were filtered in order to find the best matching situations (c.f. table 5 and 6). To clarify we have adjusted the statement.

> p.18, ll. 366f: As shown in figure 13, the results of the simulations show realistic data, even though they are not centrally located within the measurement points, therefore, other turbine parameters available are compared.

**RC 11/12**  *L368: "This can be seen in figures 19 and 20. ": These figures are not properly introduced, and their numbering does not match the text sequence.*

*L368: "Apparently, the rotor speed curve at the start of the peak shaver region is slightly different (c.f. figure 20). Therefore, it is only possible to compare loads at either the same rotor speed or the same wind speed.": There is no explanation given for this particular point. While the controller was not provided by the manufacturer, it was previously stated that the power curves matched perfectly.*

    **AC** Since the above comments refer to the same line, we have combined the answers here. You are right, that the figures should have been introduced more properly. The plots are mentioned at this point in the text, as it is important to know the criteria for the selection of the data intervals used in the load comparison. This selection was limited by i.a. the discrepancy in the rotor speed between the simulation and measurement data.
The differences in the rotor speed curve are shortly discussed in line 438. However, this might not be enough to address this issue. There are differences between the results of the simulations and the measurement data, even though the power curve (figure 11) matches very good. Apart from the differences that can be seen in figures 19 and 20, there are also differences in the standard deviation of the power (figure 12), the blade root bending moments (figure 14) and the load spectra (figures 15 to 17). Therefore, we do not claim that the results match perfectly. But the results show in many respects a very good agreement.
The differences that can be seen presumably stem on the one hand from the slightly different flow conditions (lower TI) and on the other hand from the modelling of the controller. The provided controller might not be a perfect match to the real turbine controller. But as we do not have access to the source code of it, we can only assume, that its aim was to reproduce the power from the wind speed and that the rotor speed behaviour does not match the real behaviour entirely.

> p.19, ll. 375f: This can be seen in figures 19 and 20 showing the measurements in Brusow. While figure 19 presents the relationship between the wind turbine power output and the rotor speed, figure 20 shows the relationship between rotor speed and wind speed. The combination of the respective values obtained from the simulations is provided by marks in these figures. Evidently, for the power output the values obtained for the simulation are within the standard deviation of the measurements that are indicated by bars. In that sense our setup seems to be successful. We point out that we did not set up our simulations in such a way that they would lead to the reproduction of the mean behaviour of the wind turbine for the specific bins of measured data. We simulated just a few selected cases within the neutral and stable range of atmospheric stability. Thus, a deviation of the turbine response from the mean behaviour in the measurements can be expected. Note that the cases simulated by us are cases with a comparatively low turbulence intensity. We do not know the details of the controller of the wind turbine, so a verification of any hypothesis why our cases show a smaller rotor speed in comparison with the mean rotor speed for the next bin of measured data is hard to verify.

**RC13** *Figures A.1, A.2: Could be more useful to switch the x-axis by another variable, such as radius or projected length. The nodes might not be equidistant for every aeroelastic code.*

    **AC** Thank you for this suggestion. We have adjusted the figures accordingly.

Minor comments

**RC14** *L33 "with comparatively simple models, like e.g. TurbSim" -> TurbSim seems to be a software, rather than a model. Consider using another word here*

    **AC** You are right, we have adjusted the relevant phrases.

> p.2, ll. 31f: The models currently used to calculate loads on entire wind turbines, like e.g. FAST (Jonkman and Buhl Jr., 2005) or Bladed (DNV GL, 2020), require wind fields as input, which are generally computed with comparatively simple tools, like e.g. TurbSim (Jonkman, 2009a). TurbSim and comparable software commonly use the Mann-Model (Mann, 1998) or the Kaimal-Model, c.f. (Kaimal et al., 1972), (IEC, 2005), to model turbulence.

**RC15** *L48 "Here, the use of an ALM, moving meshes and fluid–structure interaction (FSI) lead to very detailed results but also requires a further reduction of the computing time" -> the reviewer found this phrasing a bit cumbersome. Could it be simpler to state that the consideration of FSI implies an increase of the required computing time?.*

    **AC** Thank you for pointing this out. We have implemented your suggestion.

> p.2, ll. 48f: Here, the use of an ALM, moving meshes and fluid–structure interaction (FSI) lead to very detailed results but also implies high computational demands.

**RC16** *L56 "losing" -> should it be "loosing" instead?*

    **AC** The word "loose" is used when something is not tight, e.g. loose clothes, but "lose" is used when something is missing. As the intention of the phrase is "missing information", we think that the correct spelling is "losing".

**RC17** *L135 " that occur at the blades" -> experienced by the blades?*

    **AC** We have adjusted the sentence.

> p.5, ll. 137f: In general, the forces acting on the blades are calculated based on the wind speed that is present at the blade position, i.e. the positions in the rotor plane.